**Rapid reduction in ecosystem productivity caused by flash drought based on**
**decade-long FLUXNET observations**
Miao Zhang[a,c], Xing Yuan[b*]
[a]Key Laboratory of Regional Climate-Environment for Temperate East Asia
(RCE-TEA), Institute of Atmospheric Physics, Chinese Academy of Sciences, Beijing
100029, China
[b]School of Hydrology and Water Resources, Nanjing University of Information
Science and Technology, Nanjing 210044, China
[c]College of Earth and Planetary Sciences, University of Chinese Academy of Sciences,
Beijing 100049, China
Submitted May 7, 2020
Revised October 19, 2020

*Corresponding author address: Xing Yuan, School of Hydrology and Water Resources, Nanjing University of Information Science and Technology, Nanjing 210044, China E-mail: xyuan@nuist.edu.cn

**Abstract.** Flash drought is characterized by its rapid onset and arouses wide concerns due to its devastating impacts on the environment and society without sufficient early warnings. The increasing frequency of soil moisture flash drought in a warming climate highlights the importance of understanding its impact on terrestrial ecosystems. Previous studies investigated the vegetation dynamics during several extreme cases of flash drought, but there is no quantitative assessment on how fast the carbon fluxes respond to flash drought based on decade-long records with different climates and vegetation conditions. Here we identify soil moisture flash drought events by considering decline rate of soil moisture and the drought persistency, and detect the response of ecosystem carbon and water fluxes to soil moisture flash drought during its onset and recovery stages based on observations at 29 FLUXNET stations from croplands to forests. Corresponding to the sharp decline in soil moisture and higher VPD, gross primary productivity (GPP) drops below its normal conditions in the first 16 days and reduces to its minimum within 24 days for more than 50% of the 151 identified flash drought events, and savannas show highest sensitivity to flash drought. Water use efficiency increases for forests but decreases for cropland and savanna during the recovery stage of flash droughts. These results demonstrate the rapid responses of vegetation productivity and resistance of forest ecosystems to flash drought.

**Keywords:** Flash drought; GPP; Soil moisture; Water use efficiency; FLUXNET

## 1. Introduction

Terrestrial ecosystems play a key role in the global carbon cycle and absorb about 30% of anthropogenic carbon dioxide emissions during the past five decades (Le Quéré et al., 2018). With more climate extremes (e.g. droughts, heat waves) in a warming climate, the rate of future land carbon uptake is highly uncertain regardless of the fertilization effect of rising atmospheric carbon dioxide (Green et al., 2019; Reichstein et al., 2013; Xu et al., 2019). Terrestrial ecosystems can even turn to carbon source during extreme drought events (Ciais et al., 2005). Record-breaking drought events have caused enormous reduction of the ecosystem gross primary productivity (GPP), such as the European 2003 drought (Ciais et al., 2005; Reichstein et al., 2007), USA 2012 drought (Wolf et al., 2016), China 2013 drought (Xie et al., 2016; Yuan et al., 2016), Southern Africa 2015/16 drought (Yuan et al., 2017) and Australia Millennium drought (Banerjee et al., 2013). The 2012 summertime drought in USA was classified as flash drought with rapid intensification and insufficient early warning, which caused 26% reduction in crop yield (Hoerling et al., 2014; Otkin et al., 2016). Flash drought may only need several weeks to develop into its maximum intensity, and the rapid onset distinguishes it from traditional drought which is assumed to be a slowly evolving climate phenomenon taking several months or even years to develop (Otkin et al., 2018). Several extreme flash droughts would ultimately propagate into long-term droughts due to persistent precipitation deficits, e.g., 2012 flash drought over the USA Midwest Plain (Basara et al., 2019). Flash drought has aroused wide concerns for its unusually rapid development and detrimental effects

(Basara et al., 2019; Christian et al., 2019; Ford & Labosier, 2017; Nguyen et al.,
2019; Otkin et al., 2018a; Otkin et al., 2018b; Wang and Yuan, 2018; Yuan et al., 2015;
Yuan et al., 2017; Yuan et al., 2019b). Despite the increasing occurrence and clear
ecological impacts of flash droughts, our understanding of their impacts on carbon
uptake in terrestrial ecosystems remains incomplete.

Previous studies mainly focused on the response of vegetation to long-term

droughts, and found that the response time ranged from several months to years
through correlation analysis (Vicente-Serrano et al., 2013; Xu et al., 2018). The
response time of vegetation to flash droughts might be different, which requires
further investigation for quantification. Recent studies assessed the impact of flash
drought on vegetation including the 2012 central USA flash drought and the 2016 and
2017 northern USA flash drought. For instance, Otkin et al. (2016) used the
evaporative stress index (ESI) to detect the onset of the 2012 central USA flash
drought, and found the decline in ESI preceded the drought according to the United
States Drought Monitor (Svoboda et al., 2002). He et al. (2019) assessed the impacts
of the 2017 northern USA flash drought (which also impacted parts of southern
Canada) on vegetation productivity based on GOME-2 solar-induced fluorescence
(SIF) and satellite-based evapotranspiration in the US Northern plains. Otkin et al.
(2019) examined the evolution of vegetation conditions using LAI from MODIS
during the 2015 flash drought over the South-Central United States and found that the
LAI decreased after the decline of soil moisture. Besides, the 2016 flash drought over
U.S. northern plains also decreased agricultural production (Otkin et al., 2018b).

79 However, previous impact studies only focused on a few extreme flash drought cases

80 without explicit definition of flash drought events. As the baseline climate is changing

81 (Yuan et al., 2019b), it is necessary to systematically investigate the response of

82 terrestrial carbon and water fluxes to flash drought events based on long-term records

83 rather than one or two extreme cases.

84  In fact, there are numerous studies on the influence of drought on ecosystem

85 productivity (Ciais et al., 2005; Stocker et al., 2018; Stocker et al., 2019). It is found

86 that understanding the coupling of water-carbon fluxes during drought is the key to

87 revealing the adaptation and response mechanisms of vegetation to water stress

88 (Boese et al., 2019; Nelson et al., 2018). Water use efficiency (WUE) is the metric for

89 understanding the trade-off between carbon assimilation and water loss through

90 transpiration (Beer et al., 2009; Cowan and Farquhar, 1977; Zhou et al., 2014, 2015),

91 and it is influenced by environmental factors including atmospheric dryness and soil

92 moisture limitations (Boese et al., 2019). Although WUE has been widely studied for

93 seasonal to decadal droughts, few studies have investigated WUE during flash

94 droughts that usually occur at sub-seasonal time scale (Xie et al., 2016; Zhang et al.,

95 2019).

96  In this paper, we address the ecological impact of soil moisture flash droughts

97 through analyzing FLUXNET decade-long observations of $CO_2$ and water fluxes.

98 Here we consider not only the rapid onset stage of soil moisture flash droughts but

99 also the recovery stage to assess the ecological impacts. The ecological responses to

100 water stress vary under different ecosystems and drought characteristics, and the focus

on the soil moisture flash droughts would detect the breakdown of ecosystem
functioning of photosynthesis. The specific goals are to (1) examine the response of
carbon and water fluxes to soil moisture flash droughts from the onset to the recovery
stages, and (2) investigate how WUE changes during soil moisture flash drought for
different ecosystems. The methodology proposed by Yuan et al. (2019b) enables the
analysis of the flash drought with characteristics of duration, frequency, and intensity
in the historical observations. All the flash drought events occurred at the FLUXNET
stations are selected to investigate the response of carbon fluxes and WUE. More than
10-year records of soil moisture, carbon and water fluxes are available (Baldocchi et
al., 2002), which makes it possible to assess the response of vegetation to flash
droughts by considering different climates and ecosystem conditions.
**2. Data and Methods**
**2.1 Data**
FLUXNET2015 provides daily hydrometeorological variables including precipitation,
temperature, saturation vapor pressure deficit (VPD), soil moisture (sm), shortwave
radiation (SW), evapotranspiration (ET) inferred from latent heat, and carbon fluxes
including GPP and net ecosystem productivity (NEP). We use GPP data based on
night-time partitioning method (GPP_NT_VUT_REF). Considering most sites only
measure the surface soil moisture, here we use daily soil moisture measurements
mainly at the depth of 5-10 cm averaged from half-hourly data. Soil moisture
observations are usually averaged over multiple sensors including time domain
reflectometer (TDR), frequency domain reflectometer (FDR), and water content
reflectometer etc. However, the older devices may be replaced with newer devices at
certain sites, which may decrease the stability of long-term soil moisture observations
and the average observation error of soil moisture is $\pm$ 2%. All daily
hydrometeorological variables and carbon fluxes are summed to 8-day time scale to
study the flash drought impact. There are 34 sites from FLUXNET 2015 dataset
(Table 1) consisting of 8 vegetation types, where the periods of observations are no
less than 10 years ranging from 1996 to 2014, and the rates of missing data are lower
than 5%. Here we only select the FLUXNET observations including 12 evergreen
needleleaf forest sites (ENF), 5 deciduous broadleaf forests (DBF), 6 crop sites
(CROP; 5 rain-fed sites and 1 irrigated site), 3 mixed forests (MF), and 3 savannas
(SAV). The sites for grasslands, evergreen broadleaf forests, and shrublands are
excluded because there are less than 10 soil moisture flash drought events. The
vegetation classification is according to International Geosphere-Biosphere Program
(IGBP; Belward et al., 1999), where MF is dominated by neither deciduous nor
evergreen tree type with tree cover larger than 60% and the land tree cover is 10-30%
for SAV. The detailed information is listed in Table 1.
**2.2 Methods**
**2.2.1   Definition of soil moisture flash drought events**
The definition of soil moisture flash drought should account for both its rapid
intensification and the drought conditions (Otkin et al., 2018a; Yuan et al., 2019b).
Here we used soil moisture percentile to identify soil moisture flash drought
according to Yuan et al. (2019b) and Ford et al. (2017). Figure 1 shows the procedure
for soil moisture flash drought identification, including five criteria to identify the
rapid onset and recovery stages of soil moisture flash drought. 1) Soil moisture flash
drought starts at the middle day of the 8-day period when the 8-day mean soil
moisture is less than the $40^{th}$ percentile, and the 8-day mean soil moisture prior to the
starting time should be higher than $40^{th}$ percentile to ensure the transition from a
non-drought condition. 2) The mean decreasing rate of 8-day mean soil moisture
percentile should be no less than 5% per 8 days to address the rapid drought
intensification. 3) The 8-day mean soil moisture after the rapid decline should be less
than 20% in percentile, and the period from the beginning to the end of the rapid
decline is regarded as the onset stage of soil moisture flash drought (those within red
dashed line in Figure 1). 4) If the mean decreasing rate is less than 5% in percentile or
the soil moisture percentile starts to increase, the soil moisture flash drought enters
into the "recovery" stage, and the soil moisture flash drought event (as well as the
recovery stage) ends when soil moisture recovers to above $20^{th}$ percentile (those
within blue dashed line in Figure 1). The recovery stage is also crucial to assess the
impact of soil moisture flash drought (Yuan et al., 2019b). 5) The minimum duration
of a flash drought event is 24 days to exclude those dry spells that last for a too short
period to cause any impacts.

At least decade-long observations of 8-day mean soil moisture are used to

calculate soil moisture percentile with a moving window of 8-day before and 8-day
after the target 8-day, resulting in at least 30 samples for deriving the cumulative
distribution function of soil moisture before calculating percentiles. Besides, the target
8-day soil moisture percentiles are only based on the target 8-day soil moisture in the
context of the expanded samples. For example, the soil moisture percentile of June
22$^{nd}$ in 1998 is calculated by firstly ranking June 14$^{th}$, June 22$^{nd}$, and June 30$^{th}$ soil
moisture in all historical years (N samples) from lowest to highest, identifying the
rank of soil moisture of June 22$^{nd}$, 1998 (e.g., M), and obtaining the percentile as
M/N*100. We focus on growing seasons during April-September for sites in the North
Hemisphere and October-March for sites in the South Hemisphere.

**2.2.2   Response time of GPP to soil moisture flash drought**

Drought has a large influence on ecosystem productivity through altering the plant
photosynthesis and ecosystem respiration (Beer et al., 2010; Green et al., 2019;
Heimann & Reichstein, 2008; Stocker et al., 2018). GPP dominates the global
terrestrial carbon sink and it would decrease due to stomatal closure and non-stomatal
limitations like reduced carboxylation rate and reduced active leaf area index (de la
Motte et al., 2019) under water stress. The negative anomalies of GPP during soil
moisture flash drought are considered as the onset of ecological response. Here, we
use two response time indices to investigate the relationship between soil moisture
flash drought and ecological drought (Crausbay et al., 2017; Niu et al., 2018; Song et
al., 2018; Vicente-Serrano et al., 2013): 1) the response time of the first occurrence
(RT) of negative standardized GPP anomaly (SGPPA=$\frac{GPP-\mu_{GPP}}{\sigma_{GPP}}$, where $\mu_{GPP}$ and
$\sigma_{GPP}$ are mean and standard deviation of the time series of GPP at the same dates as
the target 8-days for all years, which can remove the influence of seasonality. For
instance, all Apr 1-8 during 1996-2014 would have a $\mu_{GPP}$ and a $\sigma_{GPP}$ based on a
climatology same as soil moisture percentile calculation which consists of March
24-31, Apr 1-8, and Apr 9-16 in all years, and Apr 9-16 would have another $\mu_{GPP}$
and another $\sigma_{GPP}$, and so on), which is the lag time between the start of flash drought
and the time when SGPPA becomes negative during flash drought period; and 2) the
response time of occurrence of minimum SGPPA (RTmin), which is the lag time
between the start of flash drought and the time when SGPPA decreases to its
minimum values during the flash drought period. If the response time is 8 days for the
first occurrence of negative SGPPA, it means that the response of GPP starts at the
beginning of flash drought (the first time step of flash drought). Considering flash
drought is identified through surface soil moisture due to the availability of
FLUXNET data, vegetation with deeper roots may obtain water in deep soil and
remain healthy during flash drought. The roots vary among different vegetation types
and forests are assumed to have deeper roots than grasslands, which may influence the
response to soil moisture flash droughts.
**2.2.3   Water use efficiency**
Carbon assimilation and transpiration are coupled by stomates, and plants face a
tradeoff between carbon uptake through photosynthesis and water loss through
transpiration under the influence of water and energy availability (Boese et al., 2019;
Gentine et al., 2019; Huang et al., 2016; Nelson et al., 2018). WUE can be used to
quantify the trade-off between carbon and water cycles, and is defined as the
assimilated amount of carbon per unit of water loss (Peters et al., 2018). At the
ecosystem scale, WUE is the ratio of GPP over ET (Cowan and Farquhar, 1977).
Drought would cause stomatal closure and non-stomatal adjustments in biochemical
functions thus altering the coupling between GPP and ET. Underlying WUE (uWUE)
is calculated as $GPP \times \sqrt{VPD}/ET$ considering the nonlinear relationship between
GPP, VPD and ET (Zhou et al., 2014). uWUE is supposed to reflect the relationship of
photosynthesis-transpiration via stomatal conductance at the ecosystem level by
considering the effect of VPD on WUE (Beer et al., 2009; Boese et al., 2019; Zhou et
al., 2014, 2015). WUE varies under the influence of VPD on canopy conductance
(Beer et al., 2009; Tang et al., 2006), whereas uWUE is considered to remove this
effect and be more directly linked with the relationship between environmental
conditions (e.g., soil moisture) and plant conditions (e.g., carboxylation rate; Lu et al.,
2018). The standardized anomalies of WUE and uWUE are calculated the same as
SGPPA, where different sites have different mean values and standard deviations for
different target 8-days to remove the spatial and temporal inhomogeneity.
**2.2.4   The relations between meteorological conditions and GPP**
Considering the compound impacts of temperature, radiation, VPD and soil
moisture on vegetation photosynthesis, the partial correlation is used to investigate the
relationship between GPP and each climate factor, with the other 3 climate factors as
control variables as follows:
$$r_{ij(m_1,m_2\ldots m_n)} = \frac{r_{ij(m_1,\ldots m_{n-1})} - r_{im_n(m_1,\ldots m_{n-1})} r_{jm_n(m_1,\ldots m_{n-1})}}{\sqrt{(1-r^2_{in(m_1,\ldots m_{n-1})})(1-r^2_{jn(m_1,\ldots m_{n-1})})}}$$
(1)

where $i$ represents GPP, $j$ represents the target meteorological variables and
$m_1, m_{2\ldots}$ and $m_n$ represent the control meteorological variables. $r_{ij(m_1,m_2\ldots m_n)}$ is the
partial correlation coefficient between $i$ and $j$, and $r_{ij(m_1,\ldots m_{n-1})}$, $r_{im_n(m_1,\ldots m_{n-1})}$ and
$r_{jm_n(m_1,...m_{n-1})}$ are partial correlation coefficients between $i$ and $j$, $i$ and $m_n$, $j$ and
$m_n$ respectively under control of $m_1, m_{2...} and\ m_{n-1}$.

**3. Results**

**3.1 Identification of flash drought events at FLUXNET stations**

Based on FLUXNET data, we have identified 151 soil moisture flash drought events with durations longer than or equal to 24 days using soil moisture observations of 371 site years. Figure 2a shows the distribution of the 29 sites with different vegetation types, which are mainly distributed over North America and Europe. The number of soil moisture flash drought ranges from 13 to 70 events among different vegetation types. There are 12 ENF sites in this study, and the number of soil moisture flash droughts for ENF (70) is the most among all the vegetation types. The duration for flash drought events ranges from 24 days to several months. In some extreme cases, the flash droughts would develop into long-term droughts without enough rainfall to alleviate drought conditions. Mean durations of soil moisture flash droughts for different vegetation types range from around 30 days to 50 days (Figure 2c).

Figure 3 shows the meteorological conditions during different stages of soil moisture flash drought including the standardized anomalies of temperature, precipitation, VPD, and shortwave radiation and soil moisture percentiles. Here the onset and recovery stages of flash droughts refer to certain periods characterized by the soil moisture decline rates. The standardized anomalies of temperature, precipitation, VPD, and shortwave and soil moisture percentiles are composited to show the meteorological conditions during different stages of flash droughts. The

onset stage of soil moisture flash droughts mainly refers to the rapid intensification,
and the flash droughts may or may not develop into long-term droughts depending on
the deficits in precipitation. There is a slight reduction in precipitation during 8 days
prior to soil moisture flash drought (Figure 3b). During the onset of soil moisture
flash drought, soil moisture percentiles decline rapidly from nearly 50% during 8 days
before flash drought to 18% during onset stages (Figure 3e). The rapid drying of soil
moisture is always associated with a large precipitation deficits, anomalously high
temperature and shortwave radiation and large VPD indicate increased atmospheric
dryness (Ford et al., 2017; Koster et al., 2019; Wang et al., 2016), which persist until
the recovery stage except for shortwave radiation. The soil moisture percentiles are
averaged during the onset and recovery stages and the soil moisture percentiles during
recovery stages are slightly lower than those during onset stages (Figure 3e)
considering the soil moisture is not quite dry during the early period of onset stages.
Sufficient precipitation occurs during the 8 days after soil moisture flash droughts to
relieve the drought condition and soil moisture percentiles increase from 12% during
recovery stages to 36% during 8 days after flash droughts.
**3.2 Climatological statistics of the response time of GPP to flash drought**
By analyzing all the 151 soil moisture flash drought events across 29 FLUXNET
sites, we find that negative GPP anomalies occur during 81% of the soil moisture
flash drought events. Figure 4 shows the probability distributions of the response time
of GPP to soil moisture flash drought as determined by soil moisture reductions for
the first occurrence of negative SGPPA, the minimum negative value of SGPPA and

the minimum soil moisture percentiles for different vegetation types, respectively. To

reduce the uncertainty due to small sample sizes, only the results for vegetation types

(SAV, CROP, MF, DBF, ENF) with more than 10 flash drought events are shown. For

soil moisture flash droughts from all vegetation types, the first occurrences of

negative SGPPA are concentrated during the first 24 days, and GPP starts to respond

to soil moisture flash drought within 16 days for 57% flash droughts (Figures 4a-e).

The occurrences of minimum value of SGPPA rise sharply at the beginning of soil

moisture flash drought, and reach the peak during 17-24 days, and then slow down

(Figures 4f-j), which is similar to the decline in soil moisture. Although the first

occurrences of negative SGPPA mainly occur in the onset stage, GPP would continue

to decrease in the recovery stages for 60% of soil moisture flash drought events.

Different types of vegetation including herbaceous plants and woody plants all react

to soil moisture flash drought in the early stage (Figures 4a-e). Among them, SAV

shows the fastest reaction to water stress (Figures 4a and 4f), and the RT is within 8

days for 63% events, suggesting that SAV responds concurrently with soil moisture

flash drought onset. Ultimately, 88% events for SAV show reduced vegetation

photosynthesis. The result is consistent with previous studies regarding the strong

response of semi-arid ecosystems to water availability (Gerken et al., 2019;

Vicente-Serrano et al., 2013; Zeng et al., 2018), and the decline in GPP for SAV is

related to isohydric behaviors during soil moisture drought and higher VPD, through

closing stomata to decrease water loss as transpiration and carbon assimilation

(Novick et al., 2016; Roman et al., 2015). For ENF, only 27% of soil moisture flash

droughts cause the negative SGPPA during the first 8 days. When RT is within 40
days, the cumulative frequencies range from 74% to 88% among different vegetation
types. The response frequency of RTmin and the response time of minimum soil
moisture percentiles are quite similar, although there are discrepancies among the
patterns of the response frequency for different vegetation types. The response
frequency of RTmin for SAV increases sharply during 17-24 days of soil moisture
flash droughts (Figure 4f). GPP is derived from direct eddy covariance observations
of NEP and nighttime terrestrial ecosystem respiration, and temperature-fitted
terrestrial ecosystem respiration during daytime. The response of NEP to flash
droughts shows the compound effects of vegetation photosynthesis and ecosystem
respiration. In terms of RT, the response of NEP is slower than GPP for SAV, but is
quicker for DBF and ENF (Figure 5). The discrepancies between NEP and SM in
terms of RTmin are more obvious than those between GPP and SM, and the RTmin of
NEP is much shorter than the RTmin of soil moisture especially for DBF and ENF,
which may be related to the increase of ecosystem respiration (Figures 5 i and j).

Figure 6 shows the temporal changes of SGPPA and soil moisture percentiles

during 8 days before soil moisture flash droughts and during the first 24 days of the
droughts. During 8 days before flash droughts, there is nearly no obvious decline for
SGPPA, while SAV, DBF and ENF shows small increase in GPP. The decline in
SGPPA is more significant during the first 9-24 days of soil moisture flash droughts
for different vegetation types, and SGPPA for SAV and CROP show quicker decline
even during the first 8 days of soil moisture flash droughts. The decline rates in soil
moisture are mainly concentrated within the first 16 days of flash droughts. There are
various lag times for the response of GPP to the decline in soil moisture among
different vegetation.

**3.3 The coupling between carbon and water fluxes under soil moisture stress**

Figure 7 shows the standardized anomalies of WUE and uWUE and their
components for different ecosystems during 8 days before and after soil moisture
flash droughts and the onset and recovery stages. Here, we select 81% of soil moisture
flash drought events with GPP declining down to its normal conditions to analyze the
interactions between carbon and water fluxes, while GPP during the remaining 19%
of soil moisture flash drought events may stay stable and is less influenced by drought
conditions. During 8 days before soil moisture flash drought, WUE and uWUE are
generally close to the climatology (Figure 7a) and there are no significant changes in
GPP, ET, and $ET/\sqrt{VPD}$ (Figures 7e and 7i). However, the median value of SGPPA
for SAV is positive (Figure 7e). WUE is stable during the onset stage, whereas uWUE
increases for all ecosystems except for CROP (Figure 7b). For CROP, both GPP and
ET decrease, and the decline in WUE is related with a greater reduction in GPP
relative to ET (Figure 7f and 7j). The positive anomalies of uWUE are correlated with
decrease in $ET/\sqrt{VPD}$ mainly induced by the high VPD. Increasing VPD and
deficits in soil moisture would decrease canopy conductance (Grossiord et al., 2020)
but not GPP for MF and ENF. During the onset stage, GPP and ET reduce only for
SAV, and CROP, and DBF, and the magnitudes of GPP and ET reduction are highest
for SAV. ET is close to normal conditions for MF, DBF, and ENF, thus enhancing the

drying rate of soil moisture with less precipitation supply during the onset stage. But

during recovery stage of soil moisture flash drought, GPP and ET show significant

reductions except for MF (Figures 7g and 7k), and the responses of WUE and uWUE

are different between herbaceous plants (SAV and CROP) and forests (MF, DBF, and

ENF), where WUE and uWUE decrease significantly for SAV and CROP but increase

slightly for forests (Figure 7c). The decrease in uWUE for SAV and CROP during

recovery stages indicates that SAV and CROP are likely brown due to carbon

starvation caused by the significant decrease in stomatal conductance (McDowell et

al., 2008). The decrease in GPP during recovery stage is not only related to the

reduction in canopy conductance, but also the decrease in uWUE under drought for

SAV and CROP which is possibly influenced by suppressed state of enzyme and

reduced mesophyll conductance (Flexas et al., 2012). However, the positive

anomalies of uWUE for DBF and ENF during the recover stage imply that the decline

in GPP mainly results from the stomata closure. ET starts to decrease during the

recovery stage due to the limitation of water availability, and the decreasing ET also

reflects the enhanced water stress for vegetation during the recovery stage. The

average soil moisture conditions are 12% in percentile for recovery stage but 18% for

onset stage. So, drier soil moisture in the recovery stage exacerbates ecological

response. Figure 7c also shows the higher WUE and uWUE for forests, which

indicates their higher resistance to flash drought than herbaceous plants during

recovery stage. During 8 days after flash drought, the standardized anomalies of

uWUE are still positive for forests, whereas SGPPA and ET are both lower than the

climatology for all ecosystems. The ecological negative effect would persist after the
soil moisture flash drought.

**3.4 The impact of climate factors on GPP during soil moisture flash drought**

Figure 8 shows the partial correlation coefficients between standardized
anomalies of GPP and meteorological variables and soil moisture percentiles during
different stages of soil moisture flash droughts. The correlation between climate
factors and GPP is not statistically significant during 8 days before soil moisture flash
droughts. During onset stages of soil moisture flash droughts, the partial correlation
coefficients between SGPPA and soil moisture percentiles are 0.44, 0.49 and 0.29,
respectively for SAV, CROP, and ENF ($p<0.05$). Besides, shortwave radiation is
positively correlated with SGPPA for MF, DBF, and EBF (Figure 8b) during onset
stages and the positive anomalies of shortwave radiation could partially offset the loss
of vegetation photosynthesis due to the deficits in soil moisture. SGPP is also
positively correlated with temperature during onset stages for SAV and DBF. The
partial correlation coefficients between SGPPA and VPD are -0.53 and -0.22
respectively for DBF and ENF, and the higher VPD would further decrease GPP
during onset stages. The influence of VPD on GPP is much more significant during
recovery stages and 8 days after. SGPPA is positively correlated with soil moisture
and negatively with VPD for SAV both during recovery stages and 8 days after.

**4. Discussion**

Previous studies detected the vegetation response for a few extreme drought cases
without a specific definition of flash drought from a climatological perspective (Otkin
et al., 2016; He et al., 2019). Moreover, less attention has been paid to the coupling
between carbon and water fluxes during soil moisture flash drought events. This study
investigates the response of carbon and water fluxes to soil moisture flash drought
based on decade-long FLUXNET observations during different stages of flash
droughts. The responses vary across different phases of flash drought, and different
ecosystems have different responses, which provide implications for eco-hydrological
modeling and prediction. Besides, the influence of different climate factors including
VPD and soil moisture also differs during different stages of soil moisture flash
droughts.
**4.1 The responses of carbon and water fluxes to flash droughts**
Based on 151 soil moisture flash drought events identified using soil moisture
from decade-long FLUXNET observations, the response of GPP to flash drought is
found to be quite rapid. For more than half of the 151 soil moisture flash drought
events, the GPP drops below its normal conditions during the first 16 days and
reaches its maximum reduction within 24 days. Due to the influence of ecosystem
respiration, the responses of NEP for DBF and ENF to flash droughts are much
quicker than GPP, implying that the sensitivity of ecosystem respiration is less than
that of vegetation photosynthesis (Granier et al., 2007). Eventually, 81% of soil
moisture flash drought events cause declines in GPP. During the drought period,
plants would close their stomata to minimize water loss through decreasing canopy
conductance, which in turn leads to a reduction in carbon uptake. The soil moisture
flash droughts are always accompanied by high temperature and VPD. The partial
correlation analysis shows that the increase in VPD and decrease in soil moisture both
decrease the rate of photosynthesis. High VPD further reduces canopy conductance to
minimize water loss at the cost of reducing photosynthesis during soil moisture flash
drought (Grossiord et al., 2020). The suppression of GPP and ET is more obvious for
flash drought recovery stage determined by soil moisture than the onset stage. The
discrepancy of GPP responses between different phases of soil moisture flash drought
may result from 1) soil moisture conditions which are drier during the recovery stage,
and 2) the damaged physiological functioning for specific vegetation types. The
anomalies of uWUE for ecosystems are always positive or unchanged during soil
moisture flash drought except for croplands and savannas during recovery stage. The
decrease in canopy conductance would limit photosynthetic rate, however, the
increase of uWUE may indicate adaptative regulations of ecosystem physiology
which is consistent with Beer et al. (2009). uWUE is higher than WUE during onset
stage of soil moisture flash drought, which is due to the decreased conductance under
increased VPD. However, there is no obvious difference between WUE and uWUE
during recovery stage, which indicates that photosynthesis is less sensitive to stomatal
conductance and may be more correlated with limitations of biochemical capacity
(Flexas et al., 2012; Grossiord et al., 2020). During 8 days after the soil moisture flash
drought, the anomalies of GPP and ET are still negative, indicating that the vegetation
does not recover immediately after the soil moisture flash drought. The legacy effects
of flash droughts may be related to the vegetation and climate conditions (Barnes et
al., 2016; Kannenberg et al., 2020).
This study is based on the sites that are mainly distributed over North America
and Europe. It is necessary to investigate the impact of flash drought on vegetation
over other regions with different climates and vegetation conditions. In addition, this
study used in-situ surface soil moisture at FLUXNET stations to detect vegetation
response due to the lack of soil moisture observations at deep soil layers. There would
be more significant ecological responses to flash drought identified through using
root-zone soil moisture because of its close link with vegetation dynamics. Due to the
limitation of FLUXNET soil moisture measurements, here we used soil moisture
observations mainly at the depths of 5 to 10 cm. We also analyzed the response of
GPP to flash drought identified by 0.25-degree ERA5 soil moisture reanalysis data at
the depths of 7cm and 1m. The response of GPP to flash droughts identified by
FLUXNET surface soil moisture are quite similar to those identified by ERA5 soil
moisture at the depth of 1m (not shown). There are less GPP responses to flash
droughts identified by ERA5 surface soil moisture. Although we select the ERA5 grid
cell that is closest to the FLUXNET site and use the ERA5 soil moisture data over the
same period as the FLUXNET data, we should acknowledge that the gridded ERA5
data might not be able to represent the soil moisture conditions as well as flash
droughts at in-situ scale due to strong heterogeneity of land surface. Therefore, the
in-situ surface soil moisture from FLUXNET is useful to identify flash droughts
compared with reanalysis soil moisture, although the in-situ root-zone soil moisture
would be better.
**4.2 Variation in ecological responses across vegetation types**
The responses of GPP, ET and WUE to soil moisture flash drought vary among
different vegetation types. The decline in GPP and ET only occurs across croplands
and savannas during onset stage. For most forests, the deterioration of photosynthesis
and ET appears during the recovery stage with higher WUE and uWUE. For CROP
and SAV, both WUE and uWUE decrease during the recovery stage and they may be
brown due to reduced photosynthesis. The positive anomalies of WUE and uWUE for
forests suggest that their deeper roots can obtain more water than grasslands during
flash drought. Xie et al. (2016) pointed out that WUE and uWUE for a subtropical
forest increased during the 2013 summer drought in southern China. The increased
WUE in forest sites and unchanged WUE in grasslands were also found in other
studies for spring drought (Wolf et al., 2013). In general, herbaceous plants are more
sensitive to flash drought than forests, especially for savannas. The correlation
between soil moisture and GPP is more significant for SAV, CROP, and ENF during
onset stages of flash droughts, which is consistent with the strong response to water
availability of SAV and CROP (Gerken et al., 2019). SAV is more isohydric than
forests and would reduce stomatal conductance immediately to prohibit water loss
that further exacerbates drought (Novick et al., 2016; Roman et al., 2015). However,
almost all vegetation types show high sensitivity to VPD during the recovery stage of
flash droughts.
**4.3 Potential implications for ecosystem modelling**
The study reveals the profound impact of soil moisture flash droughts on
ecosystem through analyzing eddy covariance observations. It is found that the
responses of carbon and water exchanges are quite distinguishing for forests and
herbaceous plants. For the ecosystem modeling, the response of stomatal conductance
under soil moisture stress has been addressed in previous studies (Wilson et al., 2000),
but there still exists deficiency to capture the impacts of water stress on carbon uptake
(Keenan et al., 2009), which is partly due to the different responses across species.
Incorporating physiological adaptations to drought in ecosystem modeling especially
for forests would improve the simulation of the impact of drought on the terrestrial
ecosystems.
**5. Conclusion**
This study presents how carbon and water fluxes respond to soil moisture flash
drought during 8 days before flash droughts, onset and recovery stages, and 8 days
after flash droughts through analyzing decade-long observations from FLUXNET.
Ecosystems show high sensitivity of GPP to soil moisture flash drought especially for
savannas, and GPP starts to respond to soil moisture flash droughts within 16 days for
more than half of the flash drought events under the influence of the deficit in soil
moisture and higher VPD. However, the responses of WUE and uWUE vary across
vegetation types. Positive WUE and uWUE anomalies for forests during the recovery
stage indicate the resistance to soil moisture flash drought through non-stomatal
regulations, whereas WUE and uWUE decrease for croplands and savannas during the
recovery stage. For now, the main concern about the ecological impact of soil
moisture flash drought is concentrated on the period of flash drought and the legacy
effects of flash drought are not involved. It still needs more efforts to study the

subsequent effects of soil moisture flash droughts which would contribute to assessing the accumulated ecological impacts of flash drought. Nevertheless, this study highlights the rapid response of vegetation productivity to soil moisture dynamics at sub-seasonal timescale, and different responses of water use efficiency across ecosystems during the recovery stage of soil moisture flash droughts, which complements previous studies on the sensitivity of vegetation to extreme drought at longer time scale. Understanding the response of carbon fluxes and the coupling between carbon and water fluxes to drought, especially considering the effects of climate change and human interventions (Yuan et al., 2020), might help assessing the resistance and resilience of vegetation to drought.

**Acknowledgements**

The authors thank two anonymous reviewers for their helpful comments, and thank Dr. Zhenzhong Zeng for his constructive suggestions. This work was supported by National Natural Science Foundation of China (41875105), National Key R&D Program of China (2018YFA0606002) and the Startup Foundation for Introducing Talent of NUIST. The data used in this study are all from FLUXNET 2015 (https://fluxnet.fluxdata.org/data/fluxnet2015-dataset/).

**Data availability statement**

Carbon fluxes and hydrometeorological variables from FLUXNET2015 are available through https://fluxnet.fluxdata.org/data/fluxnet2015-dataset/.

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

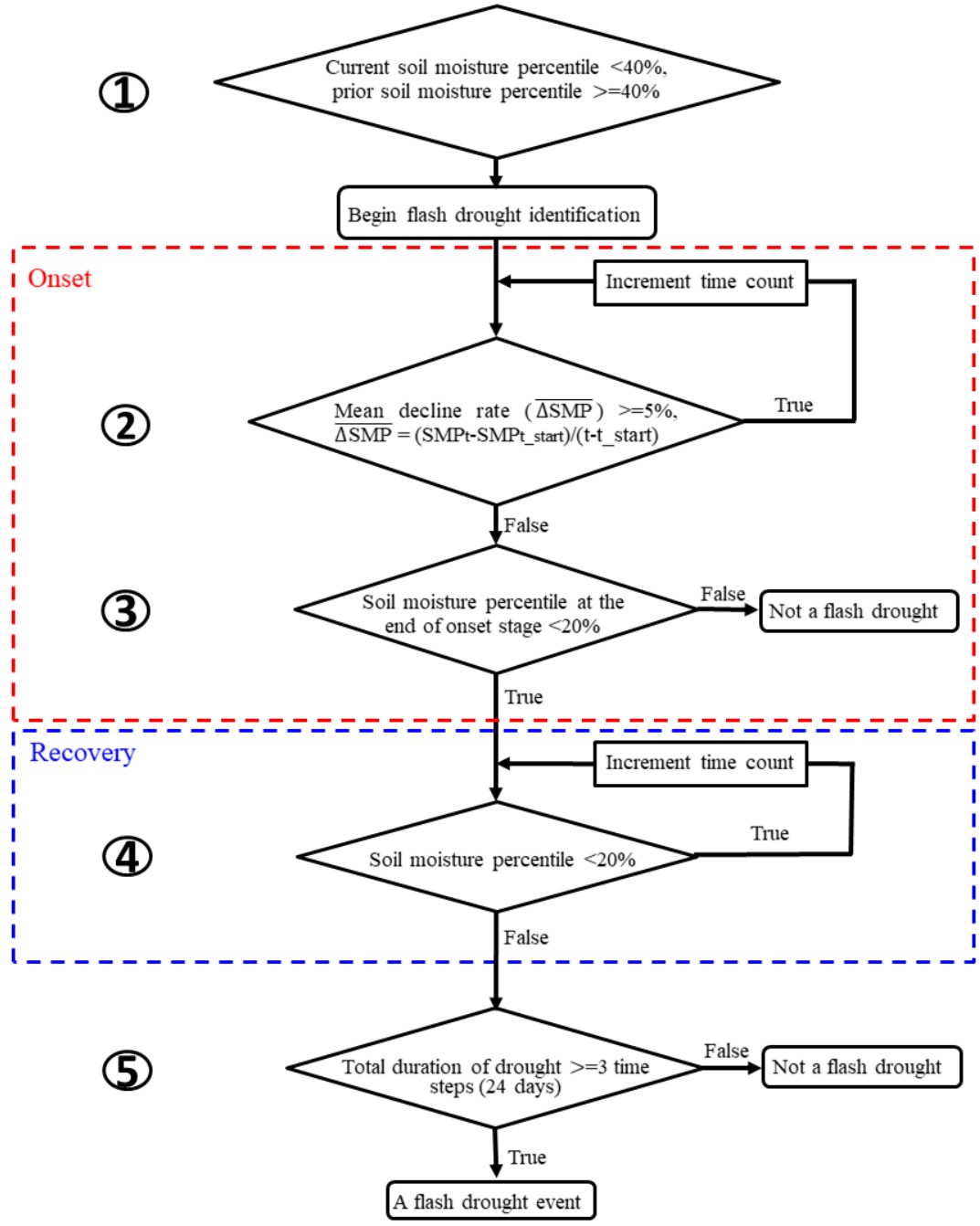


**Figure 1.** A flowchart of flash drought identification by considering soil moisture
decline rate and drought persistency.

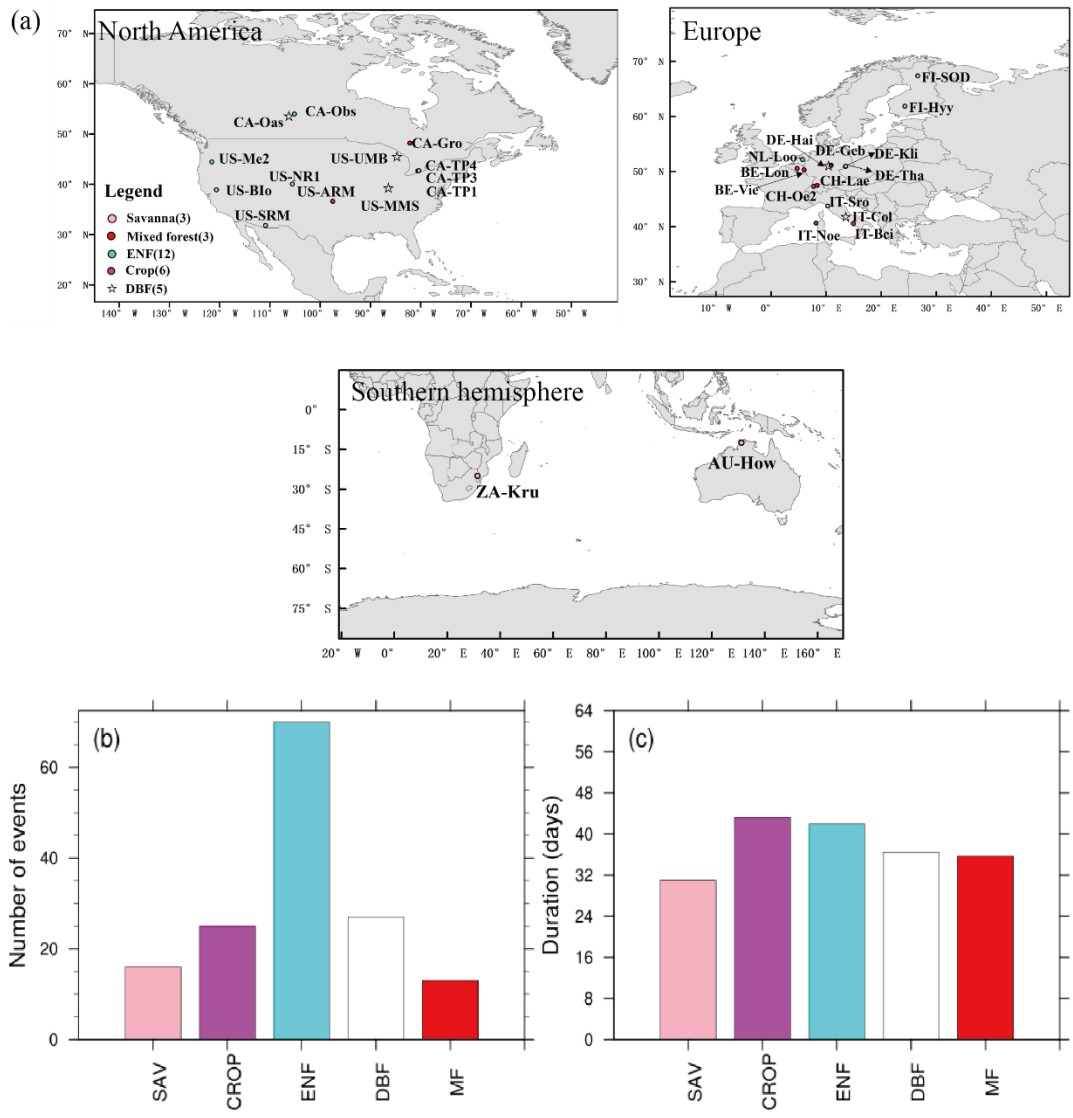


**Figure 2.** Global map of 29 FLUXNET sites used in this study (a) and flash drought

characteristics (b&c). (b) Total numbers (events) and (c) mean durations (days) of

flash drought events for each vegetation type during their corresponding periods (see

Table 1 for details). Different colors represent different vegetation types.

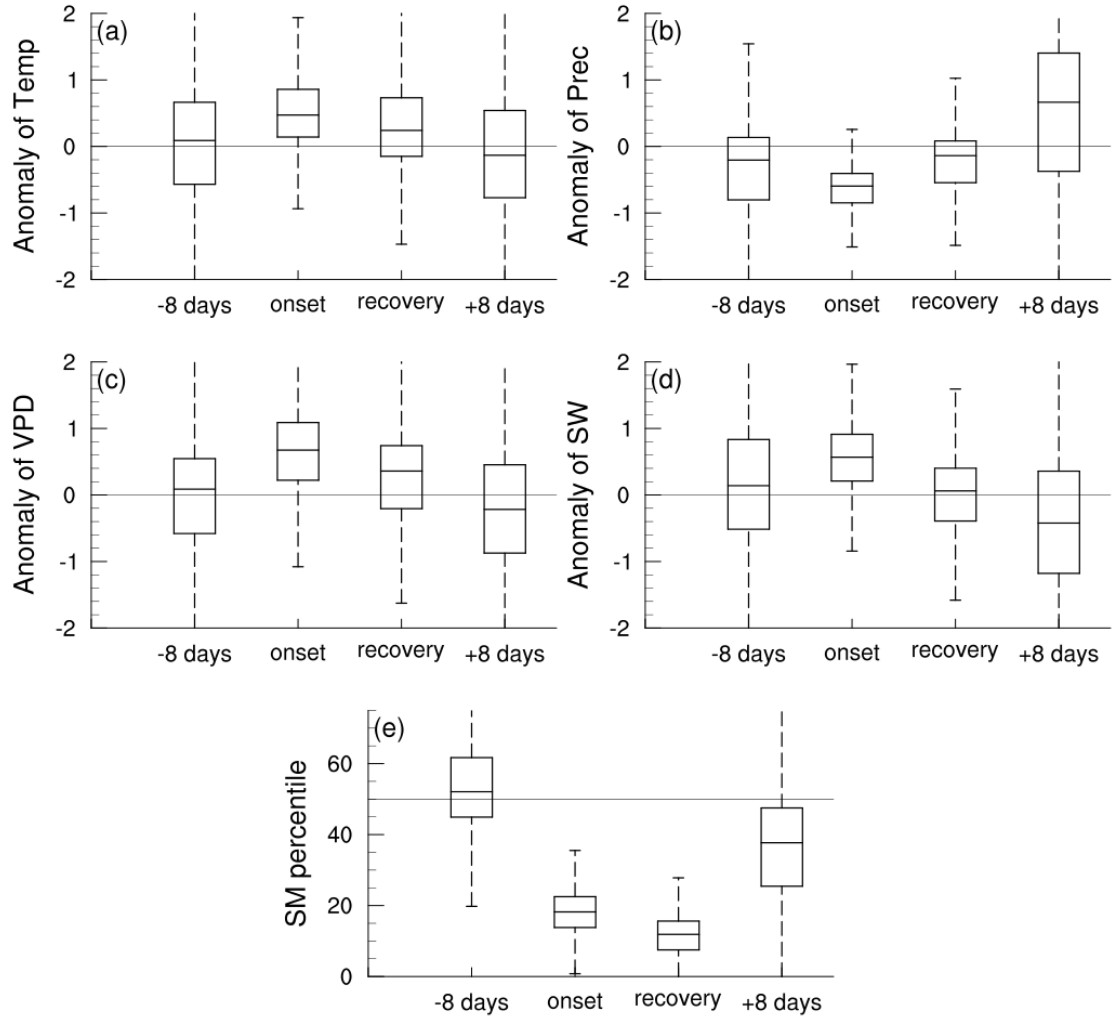


**Figure 3.** Standardized 8-day anomalies of (a) temperature, (b) precipitation, (c) VPD,
(d) short wave radiation (SW), and (e) soil moisture (SM) percentiles during 8 days
prior to flash drought onset, onset and recovery stages of flash drought, and 8 days
after flash drought.

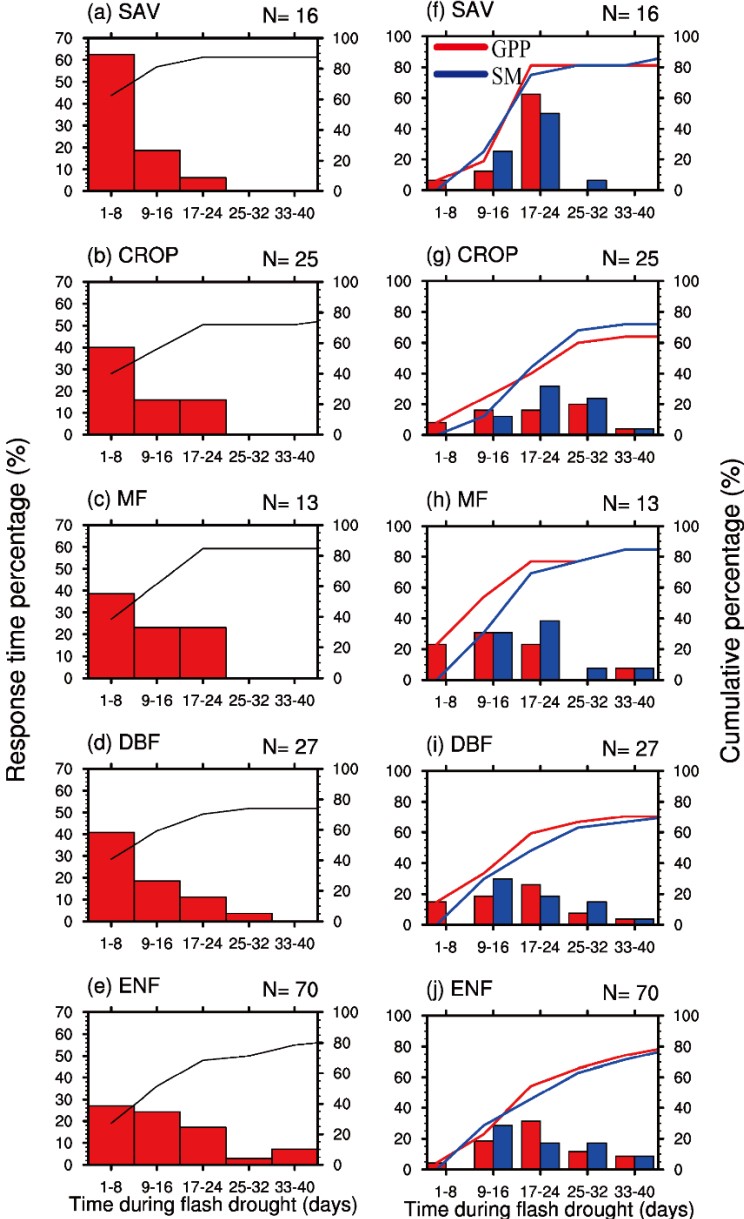


**Figure 4.** Percentage of the response time (days) of the first occurrence of negative

GPP anomaly (a-e), minimum GPP anomaly and minimum soil moisture percentile

(f-j) during soil moisture flash drought for different vegetation types. SAV: savanna,

CROP: rainfed cropland, MF: mixed forest, DBF: deciduous broadleaf forest and

ENF: evergreen needleleaf forest.

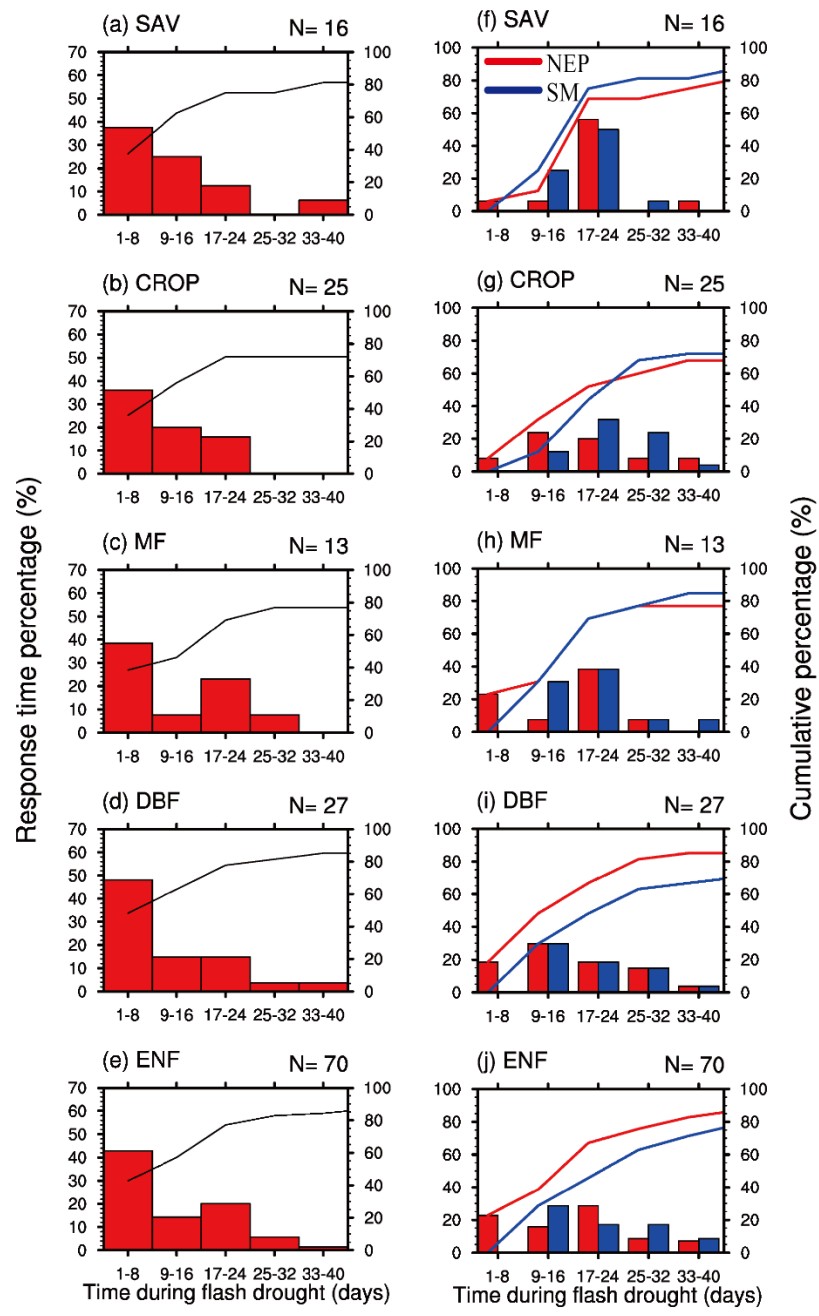

835

Figure 5. The same as Figure 4, but for net ecosystem productivity (NEP).

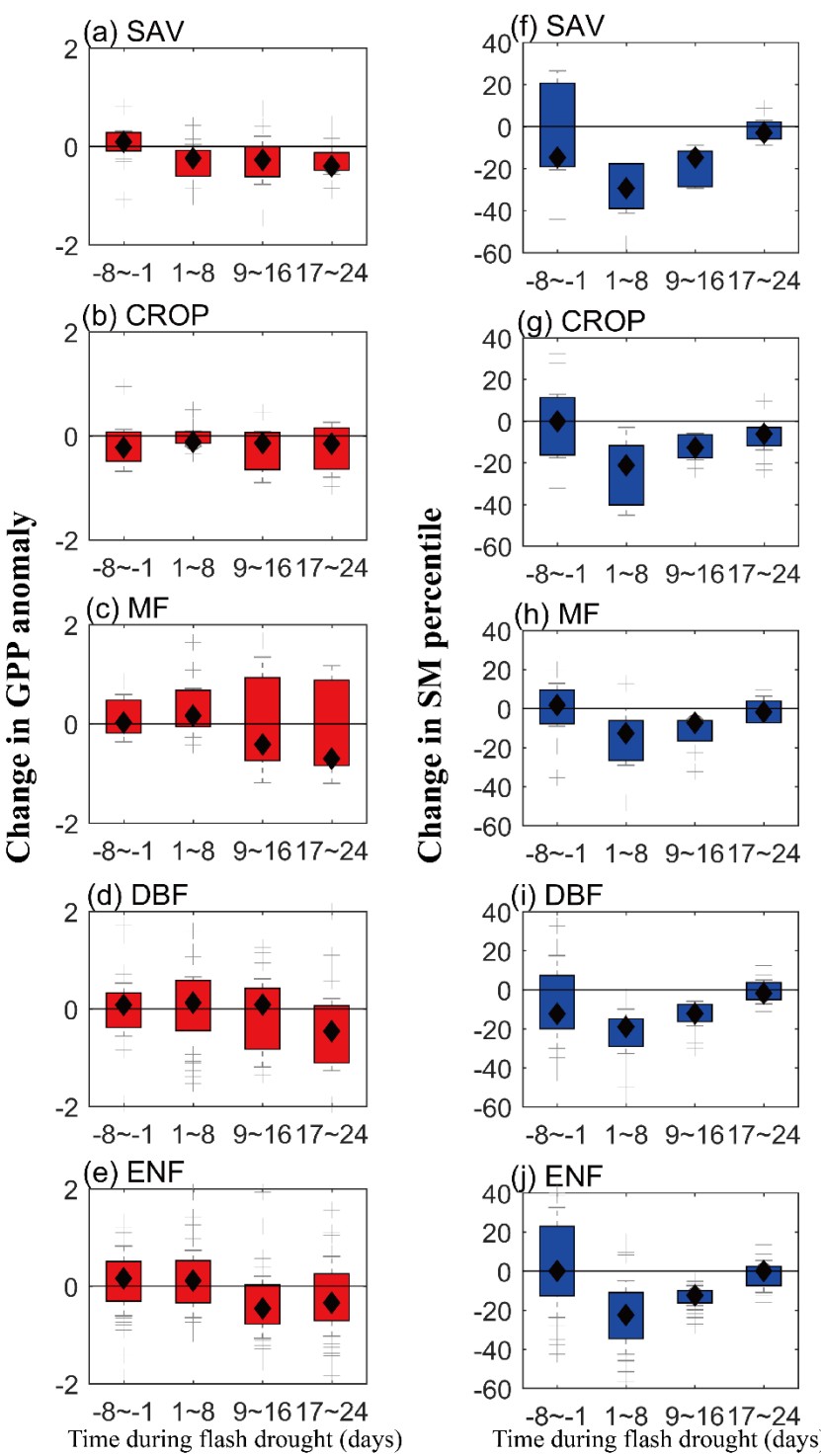

837

**Figure 6.** The temporal change rates of standardized GPP anomalies (a-e) and soil

moisture percentiles (f-j) for different vegetation types. SAV: savanna, CROP: rainfed

cropland, MF: mixed forest, DBF: deciduous broadleaf forest and ENF: evergreen

needleleaf forest.

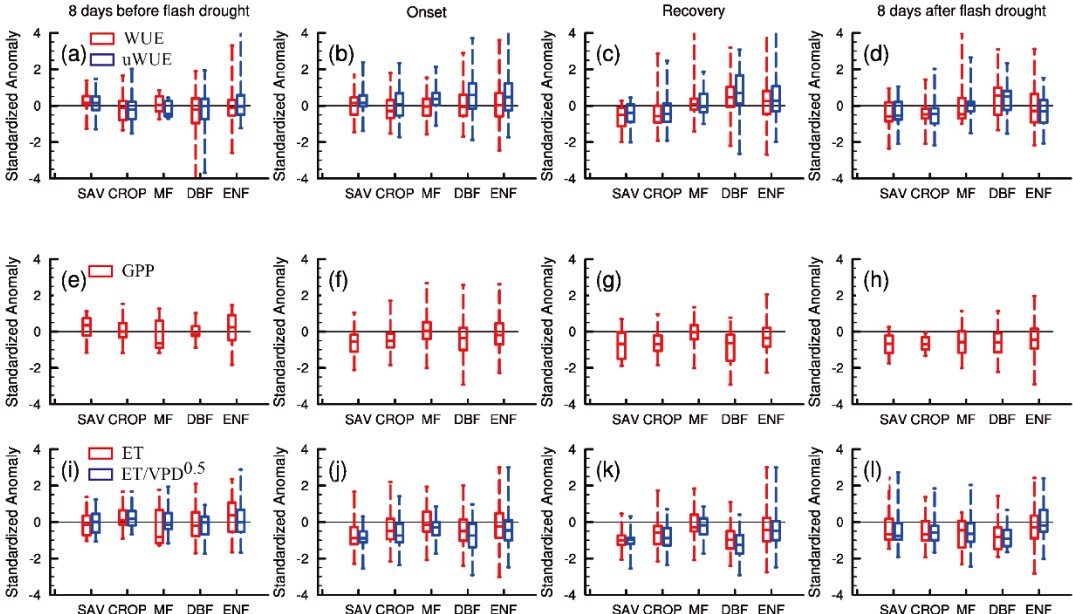

842

**Figure 7.** Standardized anomalies of water use efficiency (WUE), underlying WUE

(uWUE), GPP, ET and $ET/\sqrt{VPD}$ during 8 days before flash drought onset, onset

and recovery stages of flash drought events, and 8 days after flash drought.

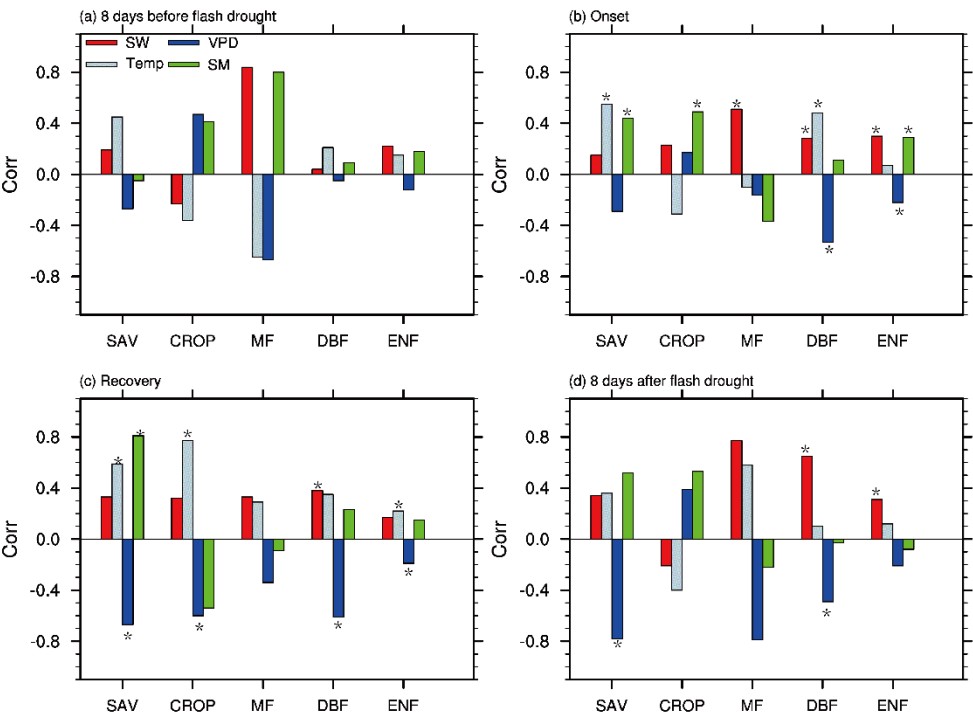

846

**Figure 8.** The partial correlation coefficients between GPP and soil moisture (SM),

shortwave radiation (SW), temperature (Temp) and vapor pressure deficit (VPD) for

different vegetation types including savannas (SAV), rain-fed croplands (CROP),

mixed forests (MF), deciduous broadleaf forests (DBF), and evergreen needleleaf

forests (ENF) during 8 days before soil moisture flash drought, onset and recovery

stages and 8 days after soil moisture flash drought. * indicates the correlation is

statistically significant at the 95% level.

**Table 1.** Locations, vegetation types and data periods of Flux Tower Sites used in this
study. WSA: woody savanna; CROP: cropland; EBF: evergreen broadleaf forests; MF:
mixed forest; DBF: deciduous broadleaf forest; ENF: evergreen needleleaf forest;
GRA: grassland; SAV: savanna.

| station | lat | lon | IGBP | period |
|---------|---------|---------|----------------|-----------|
| AU-How | -12.49 | 131.15 | WSA | 2002-2014 |
| BE-Lon | 50.55 | 4.75 | CROP-rainfed | 2004-2014 |
| BE-Vie | 50.31 | 6.00 | MF | 1997-2014 |
| CA-Gro | 48.22 | -82.16 | MF | 2004-2013 |
| CA-Oas | 53.63 | -106.20 | DBF | 1996-2010 |
| CA-Obs | 53.99 | -105.12 | ENF | 1999-2010 |
| CA-TP1 | 42.66 | -80.56 | ENF | 2002-2014 |
| CA-TP3 | 42.71 | -80.35 | ENF | 2002-2014 |
| CA-TP4 | 42.71 | -80.36 | ENF | 2002-2014 |
| CH-Lae | 47.48 | 8.37 | MF | 2005-2014 |
| CH-Oe2 | 47.29 | 7.73 | CROP-rainfed | 2004-2014 |
| DE-Geb | 51.10 | 10.91 | CROP-rainfed | 2001-2014 |
| DE-Hai | 51.08 | 10.45 | DBF | 2000-2012 |
| DE-Kli | 50.89 | 13.52 | CROP-rainfed | 2005-2014 |
| DE-Tha | 50.96 | 13.57 | ENF | 1997-2014 |
| FI-Hyy | 61.85 | 24.29 | ENF | 1997-2014 |
| FI-Sod | 67.36 | 26.64 | ENF | 2001-2014 |
| IT-Bci | 40.52 | 14.96 | CROP-irrigated | 2005-2014 |
| IT-Col | 41.85 | 13.59 | DBF | 2005-2014 |
| IT-Sro | 43.73 | 10.28 | ENF | 2000-2012 |
| NL-Loo | 52.17 | 5.74 | ENF | 1999-2013 |
| US-ARM | 36.61 | -97.49 | CROP-rainfed | 2003-2013 |
| US-Blo | 38.90 | -120.63 | ENF | 1998-2007 |
| US-Me2 | 44.45 | -121.56 | ENF | 2002-2014 |
| US-MMS | 39.32 | -86.41 | DBF | 1999-2014 |
| US-NR1 | 40.03 | -105.55 | ENF | 2002-2014 |
| US-SRM | 31.82 | -110.87 | WSA | 2004-2014 |
| US-UMB | 45.56 | -84.71 | DBF | 2002-2014 |
| ZA-Kru | -25.02 | 31.50 | SAV | 2000-2010 |