# Peer review of "1. Introduction"

_Hydrology and Earth System Sciences, 2020_

## Referee Comment (RC1) · Anonymous Referee #1 · 11 May 2020

Reviewer Recommendation: Major Revisions

General Comments:

1) Terminology – Because the definition for flash drought recovery focuses on changes in soil moisture, this framework introduces some confusion when also used to examine changes in GPP given the lag between the onset of soil moisture drought and its impact on vegetation health. For example, it is counterintuitive to refer to periods of "recovery" as those that also have substantial reductions in GPP. I think the framework used in this study is okay, but that different terminology needs to be used when referring to these periods because the "recovery" is only with respect to soil moisture conditions.

The new terminology will need to be used in the abstract, and throughout the paper. It would also help to remind the reader at various stages of the paper that "flash drought" refers to "soil moisture flash drought".

2) Definition – I think it's fine that you chose to add a maximum length threshold (lines 128-131) to the flash drought definition if you also want to solely focus on sub-seasonal drought events. However, this choice, and its impact on the resultant analysis, needs to be clearly noted in the revised text. For example, limiting flash drought duration to no more than two months means that situations where a period of rapid intensification preceded development of a longer-term drought will be excluded from the climatology because the soil moisture will not rise to greater than the 20th percentile within the chosen period of time. In fact, many of the most notable flash drought events discussed in the introduction (such as the 2012 U.S. flash drought) would presumably not be classified as flash drought with this methodology because the period of rapid intensification itself lasted for two or more months in many regions, with subsequent drought conditions lasting for many more months after that. In reality, the method used in this study only examines a subset of flash droughts, where not only must they exhibit a period of rapid intensification over 1-2 months, but then the drought conditions themselves must also be completely eliminated within another month. So, these are sub-seasonal events in their entirety. This is alluded to at lines 193-195. To reiterate, I think the methodology itself is okay, but that it needs to be clearly stated at various points of the text (abstract, methods, results, discussion, conclusions) that the goal is to look *only* at flash drought events that develop and decay over a single season, and that the method will exclude flash droughts that subsequently develop into long-term drought.

3) Section 3.3 – This section needs to be substantially revised. Given that the focus elsewhere in the paper has been to evaluate the results based on the vegetation type, it is confusing why this section primarily focuses on analyzing the results accumulated over all vegetation types in Fig. 5, before then very briefly discussing vegetationspecific results in Fig. 6. It would be much more insightful, and consistent with the rest of the paper, if you were to instead expand the existing brief analysis for each of the vegetation types into something more substantial. This would result in the removal of Fig. 5 that focuses on all of the stations in aggregate and redoing the bottom panels in Fig. 5 so that they can be added to Fig. 6 for each individual vegetation type. This will then allow you to continue to examine the time series for each vegetation type as has been done elsewhere in the paper.

Specific Comments:

4) Line 37 – Insert "future" before "land carbon uptake" in this sentence.

5) Line 58 – Please add the Svoboda et al. (2002) reference for the U.S. Drought Monitor.

6) Line 59 – This drought also impacted parts of southern Canada.

7) Line 78 – Few studies, or no studies, have investigated this parameter? If there are previous studies, please cite them here.

8) Introduction – It would also be good to cite the Otkin et al. (2018; WCAS) paper because they examined the impact of a flash drought on vegetation health across the north-central U.S.

9) Line 99 – Please add some additional information about the soil moisture sensors, such as their type, their accuracy, and how they are sited. It would also be good to know what the soil type is for each of the stations.

10) Lines 103-106 – How were these vegetation classifications determined? I think it would also be good to briefly discuss the phenological characteristics of these classifications.

11) Line 106 – Please make this sentence explicit rather than simply stating "etc". Also, this would be a good spot to point the readers to the top panel in Fig. 2 to see

the locations of these stations.

12) Lines 106-108 – Please provide some justification for why these three particular sites were chosen for the case study analyses. It would also be helpful to mention here where these three stations are located, and a brief overview of their climate characteristics. For example, are these stations located in regions that are known to frequently experience flash droughts?

13) Line 116 – Does the first day of the flash drought occur at the beginning, middle, or end of the 8-day period used to compute the mean conditions? Please clarify.

14) Figure 1 – The label between steps 2 and 3 should be "true". The box for step five should also be expanded to include "and < 2 months". Please correct these errors.

15) Please 119 – It would be good to note here that these differences are also being computed at 8-day increments to match the cadence of the 8-day mean periods.

16) Lines 123-125 – "Recovery" is imprecise here because a decrease of 4% from one period to the next does not represent recovery; instead, it simply means that the deterioration is not fast enough to meet the threshold for a flash drought used in this study. Please change this term to "stabilization", or something similar, because that will permit some degradation to still occur. Note that this only refers to the soil moisture status "stabilizing"; thus, the inconsistency with respect to the vegetation parameters (see Major Comment #1) still remains and will also need to be properly addressed.

17) Line 132 – Please change the start of this sentence to "At least decade long".

18) Lines 132-140 – It would be good to reiterate here that the percentiles themselves are still only computed over an 8-day period, but that the use of the surrounding 8-day periods are used to increase the sample size. These surrounding time periods though are certainly not completely independent, so please also comment on how much this approach does or does not increase the effective sample size when computing the percentiles.

19) Line 150 – Please add the Crausbay et al. (2017) paper in BAMS that discusses ecological drought.

20) Line 154 – You highlight an example with 19 years of data; however, most of the stations only have around 10 years of data. This is a short period for computing standard deviations. Please comment on how the short period of record will impact the anomalies and their subsequent use in this study.

21) Lines 154-157 – The example provided in this sentence implies that ecological drought always happens one period after the flash drought first develops. Is that the true intention here? If not, please clarify this sentence. I would expect there to be more than a one period lag because in many situations, the vegetation roots will extend much deeper than the 10-cm topsoil layer used in this study to identify flash droughts, thereby allowing them to remain healthy despite a rapidly drying topsoil layer. This needs to be highlighted in this section – a flash drought in the topsoil layer may not correspond to an ecological drought because of the depth of the roots.

22) Lines 150-162 – It would be helpful if each of these indices were assigned separate names to be used in the results section.

23) Line 187 – Please add "or equal to" before 24 days.

24) Line 190 – The station-level average lengths are not helpful because many of the stations only have one or two events. It would be better to show the average length over all of the stations, or for all of the stations within a particular ecosystem type. Please do this in the revised text.

25) Lines 192-193 – Is this sentence meant to imply that some stations may have multiple flash droughts because a single event is broken into two because of a rainfall event that temporally improves things? If so, please describe it as such, otherwise it is not clear what this sentence adds to the paper.

26) Line 192 – What is meant by "variability of soil moisture"? Please describe this

more clearly. Also, this really means variability of precipitation since it is the ultimate cause of the variability in soil moisture.

27) Figure 2 – The panels on this figure are difficult to read. For example, the spatial heterogeneity briefly mentioned in the text is impossible to see in the top panel because most of the stations are crammed into central Europe or North America, and it is impossible to relate the results shown in the bottom panels to the map shown in the top panel. I suggest breaking this panel into separate panels for North America, Europe, and the other four stations individually, while still taking the same amount of space as the current panel. This will allow you to zoom into all of these regions and therefore more clearly show the spatial heterogeneity.

28) Lines 204-206 – This sentence is imprecise. A decrease in ET will indeed limit the loss of soil moisture; however, it does not represent an alleviation of drought conditions. For one thing, soil moisture will still be decreasing in the absence of rainfall, albeit at a slower rate. Secondly, decreasing ET actually means that agricultural or ecological drought conditions are worsening. Please clarify this statement to account for these considerations.

29) Lines 210-211 – Please add some information describing where these stations are located, and why these events were chosen for closer analysis.

30) Figure 4 – Please change the top and bottom rows so that the precipitation and temperature anomalies can be both positive and negative, otherwise, the analysis is incomplete since only one part of the anomaly time series can be shown.

31) Line 220 – This statement is too strong because it is based on a single case study.

32) Line 228 – Is there a reference that supports this statement? The variability in the time series for this station is very similar to the other two time series shown on Fig. 4.

33) Lines 230-231 – This statement is not supported by the bottom row of Fig. 4 where the ET anomalies for this savanna station are actually less severe than those for the

forested site. Please fix this in the revised text.

34) Lines 212, 224, and 236 – It would help if you pointed the reader toward the appropriate panels on Fig. 4 in the introductions to each of these paragraphs.

35) Figure 5 – Please move the legend on panel a to panel b since that is where both of these lines are shown.

36) Line 252 – It would be good to clarify that this is "flash drought as determined by soil moisture reductions."

37) Line 279 – Why "down to its normal conditions"? I assume this is a mistake since you've already shown in the previous section that GPP anomalies become negative during a flash drought.

38) Line 284 – This ratio is reversed compared to that shown at line 172.

39) Line 288 – Again, this terminology is confusing – how can "recovery" be accompanied by "significant reductions" in GPP and ET. Those reductions show that vegetation conditions have deteriorated, not improved. This is also repeated at lines 319-320. This terminology needs to be changed to reflect that the "recovery" is only respect to soil moisture.

40) Line 315 – Please change "intensity" to "reduction".

---

## Referee Comment (RC2) · Anonymous Referee #2 · 25 May 2020

The authors present first evaluation of GPP from FLUXNET in response to flash drought. This is an important topic and this submission is timely as well as novel. At the same time, I feel that a more detailed analysis is warranted before publication.

General comments:

1) I generally think that analyzing the relationships between flash drought and GPP is very important. I am wondering though, whether this paper leaves out a large part of the story by focusing narrowly on the 30-60 days of flash drought. Similarly, there is very little analysis that looks into the underlying mechanisms of GPP besides the WUE analysis. I am wondering how temperature, global radiation, SM, and VPD, which all

affect GPP behave. For example one would expect drought to be associated with elevated temperatures. In this context, the authors stress the GPP reduction associated with drought, but several other papers have shown that GPP reduction during drought can be associated with compensation effects before and after the drought. By only focusing strictly on the drought these are being missed.

2) Similarly, the authors bin data based on onset (which should probably rather be called intensification) and recovery time as well as 8-day intervals. They present 3 examples of flash droughts in Figure 4, but it is unclear to me to what extent these are being representative and whether it makes sense to lump all drought events together like this. For example, the FI-Sod event shows fast recovery in SM, GPP, and ET (i.e. is terminated by a strong rain event), while US-SRM and IT-Col show basically no recovery of GPP and only ET recovery for IT-Col, which indicates that there is no real recovery taking place. Based on this, I would not expect to find generalizable behavior during this period. I am not sure how to resolve this in detail, but I think that a deeper dive into data and individual events is merited.

3) The discussion is falling a bit short with respect to differences between plant functional type classes. Some discussion around differences between grasslands and forests as outlined in specific comments may help here.

4) Given that FLUXNET measures NEE rather than GPP and GPP is partitioned, some discussion on this partitioning may be warranted and NEE should probably also be shown.

Specific Comments:

L99: It might be a good idea to also look into other sources of soil moisture here, as there is little standardization across FLUXNET with respect to sensor depth etc.

L101: We select 34 sites from FLUXNET where, ... > are these all sites that fit the definition from this sentence or was there further subsetting done?

L147: "The negative anomalies of GPP during flash drought are considered as the signal of ecological deterioration." > This sounds not correct to me. Water stress will reduce GPP, which is a given, but I don't think it necessarily follows that this has a lasting consequence as is implied here. It would be interesting to see to what extent do these ecosystems compensate. I.e. is there a lasting effect from a flash drought even in the annual carbon balance.

L165: "influence of water and energy conditions" > " water and energy availability?"

L189-190: "and the mean durations were from around 30 days to 60 days among FLUXNET sites" > I am a bit confused by that given that I was under the impression that droughts longer than 2 months days were excluded from the analysis. How can then mean drought length be 60 days, if that is also about the maximum possible length?

Figure 2 is problematic. I would zoom into Europe. It is also not possible to link the sites from a) to b) and c) without consulting Table 1. As a side note: the 4 Canadian ENF sites are more or less directly adjacent to each other, with 3 of them showing almost the same behavior. It may be better to only keep two of them (CA-TP4 is different (Why?))

Figure 3 and associated text: I am a bit confused about onset and recovery. Are these singe 8 day periods or do they refer to several periods. I am not sure whether this is necessarily a good way of showing this data and what is really learned here, since everything is lumped together and there is an implied time-axis, which is not consistent in itself. The temporal evolution of these events is also already well established in the literature.

Figure 4: It looks as if these sites were chosen as representative for each class, but this should be made expicit in the text. I don't particularly like the fact that anomalies are being plotted at the site level. We need to calculate ET, GPP, and SM anomalies to compare sites and establish drought, but here there is no need and it makes it harder to understand the underlying dynamics. I also think that if these sites are chosen, one

should plot all drought events (all six or so per site) and not only specifically chosen year. Also, based on this figure, I feel that onset should be renamed as intensification.

Figure 5: a) It appears if there is a quick response of GPP at the beginning of the flash drought, which one would expect simply by having high VPD, which will lead to stomata closure, but SM seems to be much less affected. It would be nice to learn whether this is really unusual or whether this GPP responses related to soil moisture reduction (drought) or VPD forcing. For example Gerken et al. 2018 (https://www.hydrol-earth-syst-sci-discuss.net/hess-2018-211/) showed that potential evapotranspiration ($\sim$ VPD) happened before the onset 2017 Norther Great Plains flash drought. It would be interesting to see whether GPP reduction also occurs before drought onset. To what extent are panels c and d necessary.

L251: "that negative GPP anomalies occur during 81%" -> if this refers to the red line in Figure 5 a/b, then this number seems inconsistent with the figure, where it is more likle 78%.

L270: "The result is consistent with the high vulnerability of vegetation in semiarid regions" > I would caution against this interpretation. Semi-arid ecosystems are highly adapted to changes in water availability and show fast response to changes in water availability (e.g. Gerken et al. 2019, 10.1038/s41612-019-0094-4). Without additional analysis, this should not be taken as a sign of degradation or vulnerability; especially since the final cumulative values are practically the same as for forests (MF, DBF, ENF). Some discussion about isohydricity, VPD may also be helpful in this context (e.g. Novick et al, 2016, 10.1038/nclimate3114, Roman et al, 2015; 10.1007/s00442-015-3380-9)

L285: "Increasing VPD and deficits in soil moisture would decrease canopy conductance" -> The fact that uWUE stays invariant shows that GPP reductions are due to canopy conductance. During recovery SAV and CROP, which are both dominated by grasses are likely brown, while forests are still green and quickly respond. This again

likes directly to different biophysical responses of forests and grasslands and isohydricity effects. These should be discussed.

L315: "Eventually, 81% of flash drought events cause negative ecological impacts on GPP." > I am not sure that a reduction in GPP is necessarily an negative impact. This depends greatly on the annual carbon balance. For example Wolf et al, 2016 (PNAS) showed that there is GPP compensation (i.e. warmer temperatures before drought causes higher initial GPP). Without looking into potential compensation effects, I feel that this statement is too harsh.

L346: "The positive anomalies of WUE and uWUE for forests show the adaptation of vegetation to flash drought from physiological perspective." > Not sure that this is true. Forests have also access to more water in the soil due to deeper roots and have invested much more in biomass. Grasslands just become dry and then recover. I think that these are different strategies rather than one being more prepared than the other.

Technical (not complete): L36: (e.g. droughtS, heat waveS)

L40: in some -> during (some is also not needed because of can)

L269: impaired -> reduced

---

## Author Comment (AC2) · 2 Jul 2020

**Response to the comments from Reviewer #2**

We are grateful to the reviewer for the constructive and careful review. The constructive suggestions have helped improved our manuscript. The reviewer's comments are italicized and our responses immediately follow.

*The authors present first evaluation of GPP from FLUXNET in response to flash drought. This is an important topic and this submission is timely as well as novel. At the same time, I feel that a more detailed analysis is warranted before publication.*

*General comments:*

*1) I generally think that analyzing the relationships between flash drought and GPP is very important. I am wondering though, whether this paper leaves out a large part of the story by focusing narrowly on the 30-60 days of flash drought. Similarly, there is very little analysis that looks into the underlying mechanisms of GPP besides the WUE analysis. I am wondering how temperature, global radiation, SM, and VPD, which all affect GPP behave. For example one would expect drought to be associated with elevated temperatures. In this context, the authors stress the GPP reduction associated with drought, but several other papers have shown that GPP reduction during drought can be associated with compensation effects before and after the drought. By only focusing strictly on the drought these are being missed.*

**Response:** Thanks for your comments. To explore the role of climate factors on GPP, we use partial correlation model to investigate the relationship between the standardized anomalies of GPP and temperature, radiation, VPD and soil moisture. Besides, we extend the study period from 8 days before flash drought to 8 days after flash drought. We have revised as follows:

**"2.2.4 The role of meteorological conditions on GPP**

Considering the compound impacts of temperature, radiation, VPD and soil moisture on vegetation photosynthesis, the partial correlation is used to investigate the relationship between GPP and each climate factor, with the other 3 climate factors as control variables as follows:

$$r_{ij(m_1,m_2\ldots m_n)} = \frac{r_{ij(m_1,\ldots m_{n-1})} - r_{im_n(m_1,\ldots m_{n-1})}r_{jm_n(m_1,\ldots m_{n-1})}}{\sqrt{(1-r_{in(m_1,\ldots m_{n-1})}^2)(1-r_{jn(m_1,\ldots m_{n-1})}^2)}} \tag{1}$$

where i represents GPP, j represents the target meteorological variables and $m_1, m_2\ldots$ and $m_n$ represent the control meteorological variables. $r_{ij(m_1,m_2\ldots m_n)}$ is the partial correlation coefficient between i and j, where $m_1, m_2\ldots$ and $m_n$ are control variables, and $r_{ij(m_1,\ldots m_{n-1})}$, $r_{im_n(m_1,\ldots m_{n-1})}$ and $r_{jm_n(m_1,\ldots m_{n-1})}$ are partial correlation coefficients between i and j, i and $m_n$, j and $m_n$ respectively under control of $m_1, m_2\ldots$ and $m_{n-1}$." (L218-230)

"**3.5 The role of climate factors on GPP during soil moisture flash drought**

Figure 8 shows the partial correlation coefficients between standardized anomalies of GPP and meteorological variables including radiation, temperature and VPD and soil moisture percentiles during different stages of soil moisture flash droughts with GPP responses. The correlation between climate factors and GPP is not statistically significant during 8 days before soil moisture flash droughts. During onset stages of soil moisture flash droughts, the partial correlation coefficients between SGPPA and soil moisture percentiles are 0.44, 0.49 and 0.29, respectively for SAV, CROP, and ENF (p<0.05). Besides, shortwave radiation is positively correlated with SGPPA for MF, DBF, and EBF (Figure 8b) during onset stages and the positive anomalies of shortwave radiation could partial offset the loss of vegetation photosynthesis due to the deficits in soil moisture. SGPP is also positively correlated with temperature during onset stages for SAV and DBF. The partial correlation coefficients between SGPPA and VPD are -0.53 and -0.22, respectively for DBF and ENF and the higher VPD would further decrease GPP during onset stages. The influence of VPD on GPP is much more significant during recovery stages and 8 days after soil moisture flash droughts. SGPPA is positively correlated with soil moisture and negatively with VPD for SAV both during recovery stages and 8 days after the soil moisture flash drought." (L422-440)

"During 8 days before soil moisture flash drought, WUE and uWUE are generally close to the climatology (Figure 7a) and there are no significant changes in GPP, ET,

and $ET/\sqrt{VPD}$ (Figures 7e and 7i). However, the median value of SGPPA for SAV is positive (Figure 7e)." (L386-389)

"During 8 days after flash drought, the standardized anomalies of uWUE are still positive, whereas SGPPA and ET are both lower than the climatology for all ecosystems. The ecological negative effect of soil moisture flash drought would persist after the flash drought due to legacy effects of drought." (L418-4421)

[Figure]

**Figure 7.** Standardized anomalies of water use efficiency (WUE), underlying WUE (uWUE), GPP, ET and $ET/\sqrt{VPD}$ during 8 days before flash drought onset, onset and recovery stages of flash drought events and 8 days after flash drought.

[Figure]

**Figure 8.** The partial correlation coefficients between GPP and soil moisture (SM), shortwave radiation (SW), temperature (temp) and vapor pressure deficit (VPD) for different vegetation types including savannas (SAV), rain-fed croplands (CROP), mixed forests (MF), deciduous broadleaf forests (DBF), and evergreen needleleaf forests (ENF) during 8 days before soil moisture flash drought, onset and recovery stages and 8 days after soil moisture flash drought. * indicates statistically significant at the 95% level.

*2) Similarly, the authors bin data based on onset (which should probably rather be called intensification) and recovery time as well as 8-day intervals. They present 3 examples of flash droughts in Figure 4, but it is unclear to me to what extent these are being representative and whether it makes sense to lump all drought events together like this. For example, the FI-Sod event shows fast recovery in SM, GPP, and ET (i.e. is terminated by a strong rain event), while US-SRM and IT-Col show basically no recovery of GPP and only ET recovery for IT-Col, which indicates that there is no real recovery taking place. Based on this, I would not expect to find generalizable behavior during this period. I am not sure how to resolve this in detail, but I think that a deeper dive into data and individual events is merited.*

**Response:** Thanks for your comments. This study emphasizes the onset of ecological response to flash droughts and three flash drought events occurring at different ecosystems are selected to show the rapid response to flash droughts, not the lasting effects of flash drought on GPP. However, it is still an important issue to assess the impacts of flash droughts and we use lagged autocorrelation models to explore the lasting effects of flash droughts on vegetation (Barnes et al., 2016) through establishing the relationship between GPP and soil moisture conditions during 8 days after flash droughts, and GPP at the end of flash droughts as follows:

$$GPP_{t+1} = b_0 + b_1 SM_{t+1} + b_2 GPP_t \tag{1}$$

where $GPP_{t+1}$ and $SM_{t+1}$ are the standardized anomalies of GPP and soil moisture percentiles during 8 days after flash droughts, and $GPP_t$ is the GPP at the end of flash droughts. $b_0$, $b_1$ and $b_2$ are empirically derived coefficients. Table R1 shows the regression coefficients of b1 and b2. The regression coefficients for soil moisture during 8 days after flash droughts is positive significantly for SAV, DBF, and ENF and the regression coefficients for GPP at the end of flash droughts are also positive for SAV and CROP (Table R1). These indicate that the antecedent vegetation conditions and soil moisture after flash droughts would influence the GPP at different ecosystems. Thus, we added the discussion about the legacy effects of flash droughts connected with climate and vegetation conditions in the revised manuscript.

**Table R1.** The regression coefficients of b1 and b2 for soil moisture during 8 days after flash droughts and the GPP at the end of flash droughts, respectively. * indicates statistically significant at the 95% level.

|  | SAV | CROP | MF | DBF | ENF |
|---|---|---|---|---|---|
| b1 | 0.009* | -0.006 | -0.006 | 0.007* | 0.001* |
| b2 | 0.82* | 0.52* | 0.11 | 0.61 | 0.56 |

"During 8 days after the soil moisture flash drought, the anomalies of GPP and ET are still negative, indicating that the vegetation does not recover immediately although the soil moisture flash drought ends. The legacy effects of flash droughts may be related to the vegetation and climate conditions (Barnes et al., 2016; Kannenberg et al.,

2020)." (L480-484)

*3) The discussion is falling a bit short with respect to differences between plant functional type classes. Some discussion around differences between grasslands and forests as outlined in specific comments may help here.*

**Response:** Thanks for your comments. We have revised the manuscript as follows:

"The correlation between soil moisture and GPP is more significant for SAV, CROP, and ENF during onset stages of flash droughts, which is consistent with the strong response to water availability of SAV and CROP (Gerken et al., 2019). SAV is more isohydric than forests and would reduce stomatal conductance immediately to prohibit water loss further exacerbating drought (Novick et al., 2016; Roman et al., 2015). However, almost all vegetation types show high sensitivity to VPD during the recovery stage of flash droughts." (L505-512)

*4)Given that FLUXNET measures NEE rather than GPP and GPP is partitioned, some discussion on this partitioning may be warranted and NEE should probably also be shown.*

**Response:** Thanks for your positive comments. We have revised our manuscript as follows:

"GPP is derived from direct eddy covariance observations of NEP and terrestrial ecosystem respiration, and the response of NEP to flash droughts shows the compound effects of vegetation photosynthesis and ecosystem respiration. In terms of RT, the response of NEP is slower than GPP for SAV, but is quicker for DBF and ENF (Figure S1). The discrepancies between NEP and SM in terms of RTmin are more obvious than those between GPP and SM, and the RTmin of NEP is much quicker than the RTmin of soil moisture especially for DBF and ENF, which may be related to the increase of ecosystem respiration (Figure S1 i and j)." (L359-367)

"Due to the influence of ecosystem respiration, the responses of NEP for DBF and ENF to flash droughts are much quicker than GPP, implying that the sensitivity of

ecosystem respiration is less than that of vegetation photosynthesis (Granier et al., 2007)." (L458-461)

[Figure]

**Figure S1.** Percentage of the response time (days) of the first occurrence of negative net ecosystem productivity (NEP) anomaly (a-e), minimum NEP anomaly and minimum soil moisture percentile (f-j) during flash drought for different vegetation types. SAV: savanna, CROP: rainfed cropland, MF: mixed forest, DBF: deciduous broadleaf forest and ENF: evergreen needleleaf forest.

*Specific Comments:*

*L99: It might be a good idea to also look into other sources of soil moisture here, as there is little standardization across FLUXNET with respect to sensor depth etc.*

**Response:** Thanks for your comments. Due to the limitation of soil moisture measurements, here we used soil moisture observations mainly at the depths of 5 to 10 cm in this study. We also analyzed the response of GPP to flash drought identified by 0.25-degree ERA5 soil moisture reanalysis data at the depths of 7cm and 1m. The response of GPP to flash droughts identified by FLUXNET surface soil moisture are quite similar to the response of GPP to flash droughts identified by ERA5 soil moisture at the depth of 1m (Figure R3). There are less GPP responses to flash droughts identified by ERA5 surface soil moisture. Although we select the ERA5 grid cell that is closest to the FLUXNET site and use the ERA5 soil moisture data over the same period as the FLUXNET data, we should acknowledge that the gridded ERA5

data might not be able to represent the soil moisture conditions as well as flash droughts at in-situ scale due to strong heterogeneity of land surface. Therefore, the in-situ soil moisture from FLUXNET is useful to identify flash droughts compared with reanalysis soil moisture, although the in-situ root-zone soil moisture would be better.

[Figure]

**Figure R3.** Response of carbon fluxes to flash droughts based on soil moisture from FLUXNET (a&b) and ERA5 at the depth of 7cm (c&d) and 1m (e&f). a, c and e are the percentages of the response time of the first occurrence of negative GPP anomaly and b, d, and f are the minimum values of GPP (red bars) and soil moisture (blue bars) during flash droughts.

*L101: We select 34 sites from FLUXNET where,…>are these all sites that fit the definition from this sentence or was there further subsetting done?*

**Response:** Thanks for your comments. We selected the study sites based on the observation period with less than 5% missing values.

*L147: "The negative anomalies of GPP during flash drought are considered as the signal of ecological deterioration.">This sounds not correct to me. Water stress will reduce GPP, which is a given, but I don't think it necessarily follows that this has a lasting consequence as implies here. It would be interesting to see to what extent do these ecosystems compensate. I.e. is there a lasting effect from a flash drought even in the annual carbon balance.*

**Response:** Thanks for your comments. We have examined the GPP response during 8 days after flash droughts in Response #1 and we have revised the manuscript as follows:

"The negative anomalies of GPP during soil moisture flash drought are considered as the onset of ecological response." (L176-178)

*L165: "influence of water and energy conditions">" water and energy availability?"*
**Response:** Revised as suggested.

*L189-190: "and the mean durations were from around 30 days to 60 days among FLUXNET sites">I am a bit confused by that given that I was under the impression that droughts longer than 2 months days were excluded from the analysis. How can then mean drought length be 60 days, if that is also about the maximum possible length?*

**Response:** Thanks for your comments. The durations of flash droughts are averaged at site level and there was only one extreme flash drought event of 56 days at IT-Noe and US-Blo.

*Figure 2 is problematic: I would zoom into Europe. It is also not possible to link the sites from a) to b) and c) without consulting Table 1. As a side note: the 4 Canadian ENF sites are more or less directly adjacent to each other, with 3 of them showing almost the same behavior. It may be better to only keep two of them (CA-TP4 is different (Why?))*

**Response:** Thanks for your comments. There are 4 Canadian ENF sites including CA-Obs, CA-TP1, CA-TP3, and CA-TP4 in this study. Although the vegetation type and climate conditions are quite similar for CA-TP1, CA-TP3, and CA-TP4, the ages of trees are different, which may influence soil moisture conditions and the ecological response to soil moisture flash droughts. We have revised Figure 2 as follows:

[Figure]

**Figure 2.** Flash drought characteristics. (a) Global map of 34 FLUXNET sites used in this study. (b&d) Total numbers (events) and (c&e) mean durations (days) of flash drought events for each site and vegetation type during their corresponding periods (see Table 1 for details). Different colors represent different vegetation types. (L)

*Figure 3 and associated text: I am a bit confused about onset and recovery. Are these singe 8 day periods or do they refer to several periods. I am not sure whether this is necessarily a good way of showing this data and what is really learned here, since*

*everything is lumped together and there is an implied time-axis, which is not consistent in itself. The temporal evolution of these events is also already well established in the literature.*

**Response:** Thanks for your comments. Here the onset and recovery stages of flash droughts refer to certain periods characterized by the soil moisture decline rates. The standardized anomalies of temperature, precipitation, VPD, and shortwave and soil moisture percentiles are composited to show the meteorological conditions during different stages of flash droughts in the revised manuscript, which is also used in Koster et al., 2019.

*Figure 4: It looks as if these sites were chosen as representative for each class, but this should be made explicit in the text. I don't particularly like the fact that anomalies are being plotted at the site level. We need to calculate ET, GPP, and SM anomalies to compare sites and establish drought, but here there is no need and it makes it harder to understand the underlying dynamics. I also think that if these sites are chosen, one should plot all drought events (all six or so per site) and not only specifically chosen year. Also, based on this figure, I feel that onset should be renamed as intensification.*

**Response:** It is not completely clear to us what the reviewer refers to here. The anomalies of ET and GPP are standardized to compare the ecological responses to flash droughts at different sites, and such analysis is quite usual like in Barriopedro et al., 2012 and Ciais et al., 2005.

"Intensification" and "onset" are quite similar to describe the development of flash droughts and the termination usually uses "onset" to describe the rapid decline in soil moisture in literature of flash droughts (Ford and Labosier, 2017; Otkin et al., 2016), thus here we use "onset" to be consistent with previous studies. Here we select three representative flash drought events from different ecosystems to reflect rapid response of GPP and ET to flash droughts. However, 81% of flash droughts would influence vegetation photosynthesis and not all flash drought events are necessary to analyze. As the reviewer suggested, we have now also introduced the locations,

vegetation types and climates of the selected sites in the revised manuscript as follows:

"Here we select FI-Sod site (26.64°E, 67.36°N), US-SRM site (110.87°W, 31.82°N) and IT-Col site (13.59°E, 41.85°N) to show the response of vegetation to flash droughts for different ecosystems and different climate. FI-Sod is with the mean annual precipitation of 500 mm yr$^{-1}$ and the mean annual temperature of -1 ℃, and it is green all the year dominated by woody vegetation of ENF. The mean annual temperature and precipitation for US-SRM are 18 ℃ and 380 mm yr$^{-1}$, respectively. US-SRM is located at SAV covered by herbaceous and other understory systems. IT-Col is dominated by DBF with leaf-on and leaf-off periods and the mean annual temperature and precipitation are 6.3 ℃ and 1180 mm yr$^{-1}$." (L119-128)

*Figure 5: a) It appears if there is a quick response of GPP at the beginning of the flash drought, which one would expect simply by having high VPD, which will lead to stomata closure, but SM seems to be much less affected. It would be nice to learn whether this is really unusual or whether this GPP responses related to soil moisture reduction (drought) or VPD forcing. For example Gerken et al. 2018 (https://www.hydrolearth-syst-sci-discuss.net/hess-2018-211/) showed that potential evapotranspiration (~VPD) happened before the onset 2017 Norther Great Plains flash drought. It would be interesting to see whether GPP reduction also occurs before drought onset. To what extent are panels c and d necessary.*

**Response:** Thanks for your positive comments. We analyzed the standardized GPP anomalies during 8 days before flash drought and there is no obvious decline in GPP. Besides, the decline in soil moisture plays a dominant role in affecting GPP during onset stages of flash droughts and the influence of higher VPD is more significant during recovery stages. Please refer to Response #1.

*L251: "that negative GPP anomalies occur during 81%"-> if this refers to the rad line in Figure 5a/b, then this number seems inconsistent with the figure, where it is more like 78%.*

**Response:** Thanks for your comments. Figure 5a/b only shows the cumulative response frequency within 1-40 days of flash droughts, which is slightly different the total response frequency. In the revised manuscript, we have deleted Figure 5 and focused on ecological responses to flash droughts for different ecosystems.

*L270: "The result is consistent with the high vulnerability of vegetation in semiarid regions" > I would caution against this interpretation. Semi-arid ecosystems are highly adapted to changes in water availability and show fast response to changes in water availability (e.g. Gerken et al. 2019, 10.1038/s41612-019-0094-4). Without additional analysis, this should not be taken as a sign of degradation or vulnerability; especially since the final cumulative values are practically the same as for forests (MF, BF, ENF). Some discussion about isohydricity, VPD may also be helpful in this context (e.g. Novick et al, 2016, 10.1038/nclimate3114, Roman et al, 2015; 10.1007/s00442015-3380-9)*

**Response:** Thanks for your comments. We have revised the manuscript as follows:

"The result is consistent with the strong response of semi-arid ecosystems to water availability (Gerken et al., 2019; Vicente-Serrano et al., 2013; Zeng et al., 2018) and the decline in GPP for SAV is more related to isohydric behaviors during soil moisture drought and higher VPD, through closing stomata to decrease water loss as transpiration and carbon assimilation (Novick et al., 2016; Roman et al., 2015)." (L346-351)

*L285: "Increasing VPD and deficits in soil moisture would decrease canopy conductance" -> The fact that uWUE stays invariant shows that GPP reductions are due to canopy conductance. During recovery SAV and CROP, which are both dominated by grasses are likely brown, while forests are still green and quickly*

*respond. This again likes directly to different biophysical responses of forests and grasslands and isohydricity effects. These should be discussed.*

**Response:** Thanks for your constructive comments.

"The decrease in uWUE for SAV and CROP during recovery stages indicates that SAV and CROP are likely brown due to carbon starvation caused by the significant decrease in stomatal conductance (McDowell et al., 2008)." (L405-407)

"However, the positive anomalies of uWUE for DBF and ENF imply that the decline in GPP mainly results from the stomata closure." (L411-412)

*L315: "Eventually, 81% of flash drought events cause negative ecological impacts on GPP." > I am not sure that a reduction in GPP is necessarily an negative impact. This depends greatly on the annual carbon balance. For example Wolf et al, 2016 (PNAS) showed that there is GPP compensation (i.e. warmer temperatures before drought causes higher initial GPP). Without looking into potential compensation effects, I feel that this statement is too harsh.*

**Response:** Thanks for your comments. We explored the response of GPP during 8 days before and after flash droughts and their relationship with soil moisture conditions and antecedent vegetation conditions in Response #1 and #2. Besides, we have revised the manuscript as follows:

"Eventually, 81% of soil moisture flash drought events cause declines in GPP." (L483-484)

*L346: "The positive anomalies of WUE and uWUE for forests show the adaptation of vegetation to flash drought from physiological perspective." > Not sure that this is true. Forests have also access to more water in the soil due to deeper roots and have invested much more in biomass. Grasslands just become dry and then recover. I think that these are different strategies rather than one being more prepared than the other.*

**Response:** Thanks for your comments. We have revised the manuscript as follows:

"The positive anomalies of WUE and uWUE for forests may be related to the adaptation of vegetation to flash drought from physiological perspective, or the deeper roots that obtain more water." (L498-501)

*Technical (not complete): L36: (e.g. droughtS, heat waveS)*

*L40: in some -> during (some is also not needed because of can)*

*L269: impaired -> reduced*

**Response:** Revised as suggested.

References:

Barriopedro, D., Gouveia, C. M., Trigo, R. M. and Wang, L.: The 2009/10 Drought in China: Possible Causes and Impacts on Vegetation, J. Hydrometeorol., 13(4), 1251–1267, doi:10.1175/JHM-D-11-074.1, 2012.

Ciais, P., Reichstein, M., Viovy, N., Granier, A., Ogée, J., Allard, V., Aubinet, M., Buchmann, N., Bernhofer, C., Carrara, A., Chevallier, F., De Noblet, N., Friend, A. D., Friedlingstein, P., Grünwald, T., Heinesch, B., Keronen, P., Knohl, A., Krinner, G., Loustau, D., Manca, G., Matteucci, G., Miglietta, F., Ourcival, J. M., Papale, D., Pilegaard, K., Rambal, S., Seufert, G., Soussana, J. F., Sanz, M. J., Schulze, E. D., Vesala, T. and Valentini, R.: Europe-wide reduction in primary productivity caused by the heat and drought in 2003, Nature, 437(7058), 529–533, doi:10.1038/nature03972, 2005.

Ford, T. W. and Labosier, C. F.: Meteorological conditions associated with the onset of flash drought in the Eastern United States, Agric. For. Meteorol., 247(April), 414–423, doi:10.1016/j.agrformet.2017.08.031, 2017.

Koster, R. D., Schubert, S. D., Wang, H., Mahanama, S. P. and DeAngelis, A. M.: Flash Drought as Captured by Reanalysis Data: Disentangling the Contributions of Precipitation Deficit and Excess Evapotranspiration, J. Hydrometeorol., 20(6), 1241–1258, doi:10.1175/jhm-d-18-0242.1, 2019.

Otkin, J. A., Haigh, T., Mucia, A., Anderson, M. C. and Hain, C.: Comparison of Agricultural Stakeholder Survey Results and Drought Monitoring Datasets

during the 2016 U.S. Northern Plains Flash Drought, Weather. Clim. Soc., 10(4), 867–883, doi:10.1175/wcas-d-18-0051.1, 2018.

---

## Author Comment (AC1)

**Response to the comments from Reviewer #1**

We are grateful to the reviewer for the constructive and careful review. We have incorporated the comments to the extent possible. The reviewer's comments are italicized and our responses immediately follow.

*General Comments:*

*1) Terminology-Because the definition for flash drought recovery focuses on changes in soil moisture, this framework introduces some confusion when also used to examine changes in GPP given the lag between the onset of soil moisture drought and its impact on vegetation health. For example, it is counterintuitive to refer to periods of "recovery" as those that also have substantial reductions in GPP. I think the framework used in this study is okay, but that different terminology needs to be used when referring to these periods because the "recovery" is only with respect to soil moisture conditions. The new terminology will need to be used in the abstract, and throughout the paper. It would also help to remind the reader at various stages of the paper that "flash drought" refers to "soil moisture flash drought"*

**Response:** Thanks for your comments. We have revised "flash drought" as "soil moisture flash drought" throughout the paper.

*2) Definition-I think it's fine that you chose to add a maximum length threshold (lines128-131) to the flash drought definition if you also want to solely focus on sub-seasonal drought events. However, this choice, and its impact on the resultant analysis, needs to be clearly noted in the revised text. For example, limiting flash drought duration to no more than two months means that situations where a period of rapid intensification preceded development of a longer-term drought will be excluded from the climatology because the soil moisture will not rise to greater than the 20$^{th}$ percentile within the chosen period of time. In fact, many of the most notable flash drought events discussed in the introduction (such as the 2012 U.S. flash drought) would presumably not be classified as flash drought with this methodology because the period of rapid intensification itself lasted for two or more months after that. In*

*reality, the method used in this study only examines a subset of flash droughts, where not only must they exhibit a period of rapid intensification over 1-2 months, but then the drought conditions themselves must also be completely eliminated within another month. So, there are sub-seasonal events in their entirety. This is alluded to at lines 193-195. To reiterate, I think the methodology itself is okay, but that is needs to be clearly stated at various points of the text (abstract, methods, results, discussion, conclusions) that the goal is to look "only" at flash drought events that develop and decay over a single season, and that the method will exclude flash droughts that subsequently develop into long-term drought.*

**Response:** Thanks for your comments. Here we concern the rapid intensification and the recovery of soil moisture for identifying flash droughts, but soil moisture percentiles does not necessarily recover to above 20% within two months. The maximum duration of 2 months is used to focus on the sub-seasonal droughts, rather than the long-term droughts. Therefore, 2012 U.S. flash drought could be identified according to the methodology used in this study. And we have clarified in the manuscript as follows:

"For example, if the soil moisture percentile does not recover to above 20% within 2 months and develops into a long-term drought (with duration longer than 2 months), only the first 2 months are used for the analysis since we focus on the sub-seasonal time scale in this study." (L155-158)

*3) Section 3.3-This section needs to be substantially revised. Given that the focus elsewhere in the paper has been to evaluate the results based on the vegetation type, it is confusing why this section primarily focuses on analyzing the results accumulated over all vegetation types in Fig.5., before then very briefly discussing vegetation specific results in Fig.6. It would be much more insightful, and consistent with the rest of the paper, if you were to instead expand the existing briefly analysis for each of the vegetation types into something more substantial. This would result in the removal of Fig.5 that focuses on all of the stations in aggregate and redoing the bottom panels in Fig.5 so that they can be added to Fig.6 for each individual vegetation type. This will*

*then allow you to continue to examine the time series for each vegetation type as has been done elsewhere in the paper.*

**Response:** Thanks for your constructive comments. We have revised the manuscript as follows:

"Figure 5 shows the probability distributions of the response time of GPP to soil moisture flash drought as determined by soil moisture reductions for the first occurrence of negative SGPPA, the minimum negative value of SGPPA and the minimum soil moisture percentiles for different vegetation types, respectively. To reduce the uncertainty due to small sample sizes, only the results for vegetation types (SAV, CROP, MF, DBF, ENF) with more than 10 flash drought events are shown. For soil moisture flash droughts from all vegetation types, the first occurrences of negative SGPPA are concentrated during the first 24 days, and GPP starts to respond to soil moisture flash drought within 16 days for 57% flash droughts (Figures 5a-e). The occurrences of minimum value of SGPPA rise sharply at the beginning of soil moisture flash drought, and reach the peak during 17-24 days, and then slow down (Figures 5f-j), which is similar to the decline in soil moisture. Although the first occurrences of negative SGPPA mainly occur in the onset stage, GPP would continue to decrease in the recovery stages for 60% of soil moisture flash drought events. Different types of vegetation including herbaceous plants and woody plants all react to soil moisture flash drought in the early stage (Figures 5a-e). Among them, SAV shows the fastest reaction to water stress (Figures 5a and 5f), and the RT is within 8 days for 63% events, suggesting that vegetation responds concurrently with soil moisture flash drought onset. Ultimately, 88% events showing reduced vegetation photosynthesis. The result is consistent with previous studies regarding the strong response of semi-arid ecosystems to water availability (Gerken et al., 2019; Vicente-Serrano et al., 2013; Zeng et al., 2018), and the decline in GPP for SAV is related to isohydric behaviors during soil moisture drought and higher VPD, through closing stomata to decrease water loss as transpiration and carbon assimilation (Novick et al., 2016; Roman et al., 2015). For ENF, the first negative SGPPA occurs within the first 8 days for 27% of the soil moisture flash droughts. When the RT is

within 40 days, the cumulative frequency ranges from 74% to 88% among different vegetation types. The response frequency of RTmin and the response time of minimum soil moisture percentiles are quite similar, although there are discrepancies among the patterns of the response frequency for different vegetation types. The response frequency of RTmin for SAV increases sharply during 17-24 days of soil moisture flash droughts (Figure 5f). GPP is derived from direct eddy covariance observations of NEP and terrestrial ecosystem respiration and the response of NEP to flash droughts show the compound effects of vegetation photosynthesis and ecosystem respiration. In terms of RT, the response of NEP is slower than GPP for SAV, but is quicker for DBF and ENF (Figure S1). The discrepancies between NEP and SM in terms of RTmin are more obvious than those between GPP and SM, and the RTmin of NEP is much quicker than the RTmin of soil moisture especially for DBF and ENF, which may relate to the increase of ecosystem respiration (Figure S1 i and j).

Figure 6 shows that the temporal changes of SGPPA and soil percentiles during 8 days before flash droughts and the first 24 days of soil moisture flash droughts. During 8 days before flash droughts, there is nearly no obvious decline for SGPPA, and SAV and ENF show small increase in GPP. The decline in SGPPA is more significant during the first 9-24 days of soil moisture flash droughts for different vegetation types and SGPPA for SAV shows quicker decline even during the first 8 days of soil moisture flash droughts. The rapid decline rates in soil moisture are mainly concentrated within the first 16 days of flash droughts and show differences among different vegetation types during soil moisture flash drought, which are related to soil texture, vegetation cover and climates. There are various lag times between the response of GPP to the decline in soil moisture among different vegetations." (L319-377)

[Figure]

**Figure 5.** Percentage of the response time (days) of the first occurrence of negative GPP anomaly (a-e), minimum GPP anomaly and minimum soil moisture percentile (f-j) during flash drought for different vegetation types. SAV: savanna, CROP: rainfed cropland, MF: mixed forest, DBF: deciduous broadleaf forest and ENF: evergreen needleleaf forest.

[Figure]

**Figure 6.** The temporal change rates of standardized GPP anomalies (a-e) and soil moisture percentiles (f-j) for different vegetation types. SAV: savanna, CROP: rainfed cropland, MF: mixed forest, DBF: deciduous broadleaf forest and ENF: evergreen needleleaf forest.

*Specific Comments:*

*4) Line 37-Insert "future" before "land carbon uptake" in this sentence.*

*5) Line 58-Please add the Svoboda et al. (2002) reference for the U.S. Drought Monitor.*

**Response:** Revised as suggested.

*6)  Line 59-This drought also impacted parts of southern Canada.*

**Response:** Thanks for your comments. He et al. (2019) assessed the impacts on vegetation productivity of the flash drought across the U.S. Northern Great Plains, excluding Canadian Prairies.

*7)  Line 78-Few studies, or no studies have investigated this parameter? If there are previous studies, please cite them here.*

*8)  Introduction-It would also be good to cite the Otkin et al. (2018; WCAS) paper because they examined the impact of a flash drought on vegetation health across the north-central U.S.*

**Response:** Revised as suggested.

*9)  Line 99-Please add some additional information about the soil moisture sensors, such as their type, their accuracy, and how they are sited. It would also be good to know what the soil type is for each of the stations.*

Response:Thanks for your comments. We have revised the manuscript as follows:

"Considering most sites only measure the surface soil moisture, here we use daily soil moisture measurements mainly at the depth of 5-10 cm averaged from half-hourly data. Soil moisture observations are usually averaged over multiple sensors including time domain reflectometer (TDR), frequency domain reflectometer (FDR), and water content reflectometer etc. However, the older devices may be replaced with newer devices at certain sites, which may decrease the stability of long-term soil moisture observations and the average observation error of soil moisture is ±2%." (L123-130)

*10) Line 103-106 -How were these vegetation classifications determined? I think it would also be good to briefly discuss the phenological characteristics of these classifications.*

**Response:** Thanks for your comments. We have revised the manuscript as follows:

"The vegetation classification is according to International Geosphere-Biosphere Program (IGBP; Belward et al., 1999), where MF is dominated by neither deciduous nor evergreen tree types with tree cover larger than 60%, and the land tree cover is 10-30% for SAV." (L116-119)

*11) Line 106- Please make this sentence explicit rather than simply stating "etc". Also, this would be a good spot to point the readers to the top panel in Fig.2 to see the locations of these stations.*

**Response:** Thanks for your comments. We have revised the manuscript as follows:

"The FLUXNET observations include 12 evergreen needleleaf forest sites (ENF), 5 deciduous broadleaf forests (DBF), 6 crop sites (CROP; 5 rain-fed sites and 1 irrigated site), 3 mixed forests (MF), 3 savannas (SAV), 2 grasslands (GRASS), 2 evergreen broadleaf forests (EBF) and 1 shrubland (Shrub)." (L112-116)

[Figure]

**Figure 2.** Flash drought characteristics. (a) Global map of 34 FLUXNET sites used in this study. (b&d) Total numbers (events) and (c&e) mean durations (days) of flash drought events for each site and vegetation type during their corresponding periods (see Table 1 for details). Different colors represent different vegetation types. (L)

*12) Lines 106-108-Please provide some justification for why these three particular sites were chosen for the case study analyses. It would also be helpful to mention here where these three stations are located, and a brief overview of their climate*

*characteristics. For example, are there stations located in regions that are known to frequently experience flash droughts?*

**Response:** Thanks for your comments. Here the selected three sites represent different ecosystems including savanna, evergreen needleleaf forest, and deciduous broadleaf forest, which are located at high latitude and middle latitude with distinguished climates, respectively. We have revised the manuscript as follows:

"Here we select FI-Sod site (26.64°E, 67.36°N), US-SRM site (110.87°W, 31.82°N) and IT-Col site (13.59°E, 41.85°N) to show the response of vegetation to flash droughts for different ecosystems and different climate. FI-Sod is with the mean annual precipitation of 500 mm $yr^{-1}$ and the mean annual temperature of -1℃, and it is green all the year dominated by woody vegetation of ENF. The mean annual temperature and precipitation for US-SRM are 18℃ and 380 mm $yr^{-1}$, respectively. US-SRM is located at SAV covered by herbaceous and other understory systems. IT-Col is dominated by DBF with leaf-on and leaf-off periods and the mean annual temperature and precipitation are 6.3℃ and 1180 mm $yr^{-1}$." (L120-129)

*13) Line 116-Does the first day of the flash drought occur at the beginning, middle, or end of the 8-day period used to compute the mean conditions? Please clarify.*

**Response:** Thanks for your comments. We have revised the manuscript as follows:

"1) Soil moisture flash drought starts at the middle day of the 8-day period when the 8-day mean soil moisture is less than the $40^{th}$ percentile, and the 8-day mean soil moisture prior to the starting time should be higher than $40^{th}$ percentile to ensure the transition from a non-drought condition." (L138-142)

*14) Figure 1-The label between steps 2 and 3 should be "true". The box for step five should also be expanded to include "and <2 months". Please correct these errors.*

**Response:** Thanks for your comments. We have revised the manuscript as follows:

[Figure]

**Figure 1.** A flowchart of flash drought identification by considering soil moisture decline rate and drought persistency.

*15) Line 119-It would be good to note here that these differences are also being computed at 8-day increments to match the cadence of the 8-day mean periods.*

**Response:** Thanks for your comments. We have revised the manuscript as follows:

"2) The mean decreasing rate of 8-day mean soil moisture percentile should be no less than 5% per 8 days to address the rapid drought intensification." (L142-143)

*16) Lines 123-125-"Recovery" is imprecise here because a decrease of 4% from one period to the next does not represent recovery; instead, it simply means that the deterioration is not fast enough to meet the threshold for a flash drought used in this study. Please change this term to "stabilization", or something similar, because that will permit some degradation to still occur. Note that this only refers to the soil moisture status "stabilizing", thus, the inconsistency with respect to the vegetation parameters (see Major Comment#1) still remains and will also need to be properly addressed.*

**Response:** Thanks for your comments. The end of the onset stage of flash drought occurs when the **mean decline rate** is smaller than 5% in percentiles per 8 days, which would avoid such phenomenon that the soil moisture percentiles are still declining after the onset stage as much as possible. And we compared the soil moisture percentiles during recovery stages and at the ending point of onset stages and the soil moisture still declines at the rate of 2~3% in percentiles per 8 days only for 3% of flash drought events (Figure R1). Therefore, the soil moisture percentiles during the identified recovery stages increase as compared with the ending point of onset stages for most cases.

[Figure]

**Figure R1.** The frequency of soil moisture percentile changes between recovery

stages and the ending point of onset stages.

*17) Line 132-Please change the start of this sentence to "At least decade long"*

**Response:** Revised as suggested.

*18) Line 132-140-It would be good to reiterate here that the percentiles themselves are still only computed over an 8-day period, but that the use of the surrounding 8-day periods are used to increase the sample size. These surrounding time periods though are certainly not completely independent, so please also comment on how much this approach does or does not increase the effective sample size when computing the percentiles*

**Response:** Thanks for your comments. Figure R2 shows the probability density function of soil moisture at different time based on the climatology solely from the target time of all observation years (a_clim) and the climatology consisting of the target time and 8 days before and after the target time of all observation years (b_clim). The b_clim is smoother than a_clim, indicating that the extended samples would decrease the uncertainty caused by certain extreme values. We have revised the manuscript as follows:

"Besides, the target 8-day soil moisture percentiles are only based on the target 8-day soil moisture in the context of the expanded samples." (L163-164)

[Figure]

**Figure R2.** The probability density function of soil moisture at Jun 2-9, Jun 10-Jul 17, Jun 18-Jun 25, Jun 26-Jul 2 based on the climatology solely from the target time during all observation years (black lines; a_clim) and the climatology from not only the target time but also 8 days before and after the target time from all observation years (red lines; b_clim).

*19) Lines 150-Please add the Crausbay et al. (2017) paper in BAMS that discusses ecological drought.*

**Response:** Revised as suggested.

*20) Line 154-You highlight an example with 19 years of data: however, most of the stations only have around 10 years of data. This is a short period for computing standard deviations. Please comment on how the short period of record will impact the anomalies and their subsequent use in this study.*

**Response:** Thanks for your comments. Here the standardized deviation of GPP are also based on at least 30-sample climatology, which is same as that of soil moisture percentiles. We have revised the manuscript as follows:

"For instance, all Apr 1-8 during 1996-2014 would have a $\mu_{GPP}$ and a $\sigma_{GPP}$ based

on a climatology same as soil moisture percentile calculation, which consists of March 24-31, Apr 1-8, and Apr 9-16 in all years, and Apr 9-16 would have another $\mu_{GPP}$ and another $\sigma_{GPP}$" (L184-187)

*21) Lines 154-157-The example provided in this sentence implies that ecological drought always happens one period after the flash drought first develops. Is that the true intention here? If not, please clarify this sentence. I would expect there to be more than a one period lag because in many situations, the vegetation roots will extend much deeper than the 10-cm topsoil layer used in this study to identify flash droughts, thereby allowing them to remain healthy despite a rapidly drying topsoil layer. This needs to be highlighted in this section – a flash drought in the top soil layer may not correspond to an ecological drought because of the depth of the roots.*

**Response:** Thanks for your comments. We have revised the manuscript as follows: "Considering flash drought is identified through surface soil moisture, vegetation with deeper roots may obtain water in deep soil moisture and remain healthy during flash drought." (L194-196)

*22) Lines 150-162-It would be helpful if each of these indices were assigned separate names to be used in the results section.*

*23) Line 187-Please add "or equal to" before 24 days*

**Response:** Revised as suggested.

*24) Line 190-The station level average lengths are not helpful because many of the stations only have one or two events. It would be better to show the average length over all of the stations, or for all of the stations within a particular ecosystem type. Please do this in the revised text.*

**Response:** Thanks for your comments. We have revised the manuscript as suggested and Figure 2 is shown in Response #11. The manuscript has been revised as follows: "Figure 2a shows the distribution of the 34 sites with different vegetation types, which are mainly distributed over North America and Europe. The number of soil moisture

flash drought ranges from 1 to 12 events among FLUXNET sites (Figure 2b). There are 12 ENF sites in this study, and the number of soil moisture flash droughts for ENF is the most among all the vegetation types. There are less than 10 flash droughts at GRASS, Shrub, EBF (Figure 2d). Mean durations of soil moisture flash droughts range from 30 days to 56 days and the duration for Shrub is quite high due to one extreme flash drought (Figure 2e)." (L234-242)

*25) Lines 192-193-Is this sentence meant to imply that some stations may have multiple flash droughts because a single event is broken into two because of a rainfall event that temporally improves things? If so, please describe it as such, otherwise it is not clear what this sentence adds to the paper.*

**Response:** Thanks for your comments. We have deleted this sentence.

*26) Line 192-What is meant by "variability of soil moisture"? Please describe this more clearly. Also, this really means variability of precipitation since it is the ultimate cause of the variability in soil moisture.*

**Response:** Thanks for your comments. We have deleted this sentence.

*27) Figure 2-The panels on this figure are difficult to read. For example, the spatial heterogeneity briefly mentioned in the text is impossible to see in the top panel because most of the stations are crammed into central Europe or North America, and it is impossible to relate the results shown in the bottom panels to the map shown in the top panel. I suggest breaking this panel into separate panels for North America, Europe, and the other four stations individually, while still taking the same amount of space as the current panel. This will allow you to zoom into all of these regions and therefore more clearly show the spatial heterogeneity.*

**Response:** Thanks for your comments and we have revised Figure 2 as suggested. Please refer to Response #11.

*28) Lines 204-206-This sentence is imprecise. A decrease in ET will indeed limit the*

*loss of soil moisture; however, it does not represent an alleviation of drought conditions. For one thing, soil moisture will still be decreasing in the absence of rainfall, albeit at a slower rate. Secondly, decreasing ET actually means that agricultural or ecological drought conditions are worsening. Please clarify this statement to account for these considerations.*

**Response:** We have moved the analysis of ET into Section 3.4 in the revised manuscript.

*29) Lines 210-211-Please add some information describing where these stations are located, and why these events were chosen for closer analysis.*

**Response:** Thanks for your comments. We have added detailed information about the selected sites in the revised manuscript. Please refer to Response #12.

*30) Figure 4-Please change the top and bottom rows so that precipitation and temperature anomalies can be both positive and negative, otherwise, the analysis is incomplete since only one part of the anomaly time series can be shown.*

**Response:** We have revised Figure 4 as follows:

[Figure]

**Figure 4.** Time series of soil moisture percentiles (a, f, k) and standardized anomalies of temperature (b, g, l), precipitation (c, h, m), gross primary productivity (GPP; d, i, n) and evapotranspiration (ET; e, j, o) for the 2003 drought at FI-Sod station, 2004 drought at US-SRM station and 2007 drought at IT-Col station. Red lines are the time series in the target year, and black lines are the climatology (long-term mean values). The blue and pink shaded areas are the onset and recovery stages of flash drought events, respectively. (L849-858)

*31) Line 220-This statement is too strong because it is based on a single case study.*

**Response:** Thanks for your comments. We have deleted this sentence.

*32) Line 228-Is there a reference that supports this statement? The variability in the time series for this station is very similar to the other two time series shown on Fig.4.*

**Response:** Thanks for your comments. We have deleted this sentence.

*33) Lines 230-231-This statement is not supported by the bottom row of Fig. 4 where the ET anomalies for this savanna station are actually less severe than those for the forested site. Please fix this in the revised text.*

**Response:** Thanks for your comments. We have deleted this sentence.

*34) Lines 212, 224, and 236-It would help if you pointed the reader toward the appropriate panels on Fig. 4 in the introductions to each of these paragraphs.*

**Response:** Thanks for your comments. We have revised this manuscript as suggested.

*35) Figure 5-Please move the legend on panel a to panel b since that is where both these lines are shown.*

*36) Line 252-It would be good to clarify that is "flash drought as determined by soil moisture reductions"*

**Response:** Revised as suggested.

*37) Line 279-Why "down to its normal conditions"? I assume this is a mistake since you've already shown in previous section that GPP anomalies become negative during a flash drought.*

**Response:** Here negative GPP anomalies did not occur during all flash drought events and GPP responded to 81% of flash droughts in this study.

*38) Line 284-The ratio is reversed compared to that shown at line 172.*

**Response:** Here uWUE (GPP $\times \sqrt{VPD}/ET$) is partitioned into GPP and $ET/\sqrt{VPD}$, which is more direct when compared the response of vegetation photosynthesis and stomatal conductance to soil moisture flash droughts, respectively.

*39) Line 288-Again, this terminology is confusing-how can "recovery" be accompanied by "significant reductions" in GPP and ET. Those reductions show that vegetation conditions have deteriorated, not improved. This is also repeated at lines*

*319-320. This terminology needs to be changed to reflect that the "recovery" is only respect to soil moisture.*

*40) Line 315-Please change "intensify" to "reduction".*

**Response:** Revised as suggested.

---

## Author Response (AR1)

Xing Yuan
Professor
School of Hydrology and Water Resources
Nanjing University of Information Science and Technology
No.219, Ningliu Road, Nanjing 210044, Jiangsu, China
Email: xyuan@nuist.edu.cn
Tel: +86-025-58699958
https://orcid.org/0000-0001-6983-7368
July 30, 2020

Re: hess-2020-185

Dear Dr. Hildebrandt,

Regarding your decision letter on our manuscript entitled "Rapid reduction in ecosystem productivity caused by flash drought based on decade-long FLUXNET observations" (hess-2020-185), we have now carefully considered your and reviewers' comments and incorporated them into the manuscript to the extent possible. The main changes include revising the flash drought definition by dropping the 60-day duration limit to consider all flash drought cases (although it does not affect the conclusions in this study), analyzing the responses of GPP and ET during different stages of flash droughts, investigating the climate controls on GPP using partial correlation analysis, extending the discussion, and providing point-by-point responses. We hope that you find the revised manuscript and the response acceptable to *Hydrology and Earth System Sciences*. The detailed responses to the comments are attached.

We appreciate the effort you spent to process the manuscript and look forward to hearing from you soon.

Sincerely yours,

Xing Yuan

**Response to the comments from Reviewer #1**

We are grateful to the reviewer for the constructive and careful review. We have incorporated the comments to the extent possible. The reviewer's comments are italicized and our responses immediately follow.

*General Comments:*

*1) Terminology-Because the definition for flash drought recovery focuses on changes in soil moisture, this framework introduces some confusion when also used to examine changes in GPP given the lag between the onset of soil moisture drought and its impact on vegetation health. For example, it is counterintuitive to refer to periods of "recovery" as those that also have substantial reductions in GPP. I think the framework used in this study is okay, but that different terminology needs to be used when referring to these periods because the "recovery" is only with respect to soil moisture conditions. The new terminology will need to be used in the abstract, and throughout the paper. It would also help to remind the reader at various stages of the paper that "flash drought" refers to "soil moisture flash drought"*

**Response:** Yes, given the definition of flash drought in this study is based on soil moisture deficit and decline rate, the "recovery" means the recovery of soil moisture drought instead of ecological drought. There is a lagged effect of ecosystem to soil moisture drought, so the GPP recovery usually lags behind the soil moisture recovery. According to the suggestion, we have revised "flash drought" as "soil moisture flash drought" throughout the paper.

*2) Definition-I think it's fine that you chose to add a maximum length threshold (lines128-131) to the flash drought definition if you also want to solely focus on sub-seasonal drought events. However, this choice, and its impact on the resultant analysis, needs to be clearly noted in the revised text. For example, limiting flash drought duration to no more than two months means that situations where a period of rapid intensification preceded development of a longer-term drought will be excluded from the climatology because the soil moisture will not rise to greater than the 20th*

*percentile within the chosen period of time. In fact, many of the most notable flash drought events discussed in the introduction (such as the 2012 U.S. flash drought) would presumably not be classified as flash drought with this methodology because the period of rapid intensification itself lasted for two or more months after that. In reality, the method used in this study only examines a subset of flash droughts, where not only must they exhibit a period of rapid intensification over 1-2 months, but then the drought conditions themselves must also be completely eliminated within another month. So, there are sub-seasonal events in their entirety. This is alluded to at lines 193-195. To reiterate, I think the methodology itself is okay, but that is needs to be clearly stated at various points of the text (abstract, methods, results, discussion, conclusions) that the goal is to look "only" at flash drought events that develop and decay over a single season, and that the method will exclude flash droughts that subsequently develop into long-term drought.*

**Response:** Thanks for your comments. In the last version of the manuscript, we only focused on the first two months of the flash drought if it did not recover. So we actually did not remove those flash droughts with long durations, but the maximum length threshold may affect the analysis during the recovery stage and after the flash drought. To avoid the confusion, we have now removed the maximum length threshold to consider the whole evolution of flash drought events even if it lasts for more than two months. In the revised manuscript, there are 151 flash drought events, and 20 of them have durations that are longer than two months. However, the main conclusions remain the same. The changes related to the removal to maximum length threshold are as follows:

"The number of soil moisture flash drought ranges from 13 to 70 events among different vegetation types. There are 12 ENF sites in this study, and the number of soil moisture flash droughts for ENF (70) is the most among all the vegetation types. The duration for flash drought events ranges from 24 days to several months. In some extreme cases, the flash droughts would develop into long-term droughts without enough rainfall to alleviate drought conditions. Mean durations of soil moisture flash droughts for different vegetation types range from around 30 days to 50 days (Figure

2c)." (L232-241)

[Figure]

**Figure 1.** A flowchart of flash drought identification by considering soil moisture decline rate and drought persistency.

[Figure]

**Figure 2.** (a) Global maps of 29 FLUXNET sites used in this study. (b) Total numbers (events) and (c) mean durations (days) of soil moisture flash drought events for each vegetation type during their corresponding periods (see Table 1 for details). Different colors represent different vegetation types.

*3) Section 3.3-This section needs to be substantially revised. Given that the focus elsewhere in the paper has been to evaluate the results based on the vegetation type, it is confusing why this section primarily focuses on analyzing the results accumulated over all vegetation types in Fig.5., before then very briefly discussing vegetation specific results in Fig.6. It would be much more insightful, and consistent with the rest of the paper, if you were to instead expand the existing briefly analysis for each of the*

*vegetation types into something more substantial. This would result in the removal of Fig.5 that focuses on all of the stations in aggregate and redoing the bottom panels in Fig.5 so that they can be added to Fig.6 for each individual vegetation type. This will then allow you to continue to examine the time series for each vegetation type as has been done elsewhere in the paper.*

**Response:** Thanks for your constructive comments. We have reorganized the results and shown them for each vegetation type, and removed those results accumulated over all vegetation types. We have revised the manuscript as follows:

"Different types of vegetation including herbaceous plants and woody plants all react to soil moisture flash drought in the early stage (Figures 4a-e). Among them, SAV shows the fastest reaction to water stress (Figures 4a and 4f), and the RT is within 8 days for 63% events, suggesting that SAV responds concurrently with soil moisture flash drought onset. Ultimately, 88% events for SAV show reduced vegetation photosynthesis. The result is consistent with previous studies regarding the strong response of semi-arid ecosystems to water availability (Gerken et al., 2019; Vicente-Serrano et al., 2013; Zeng et al., 2018), and the decline in GPP for SAV is related to isohydric behaviors during soil moisture drought and higher VPD, through closing stomata to decrease water loss as transpiration and carbon assimilation (Novick et al., 2016; Roman et al., 2015). For ENF, only 27% of soil moisture flash droughts cause the negative SGPPA during the first 8 days. When RT is within 40 days, the cumulative frequencies range from 74% to 88% among different vegetation types. The response frequency of RTmin and the response time of minimum soil moisture percentiles are quite similar, although there are discrepancies among the patterns of the response frequency for different vegetation types. The response frequency of RTmin for SAV increases sharply during 17-24 days of soil moisture flash droughts (Figure 4f). GPP is derived from direct eddy covariance observations of NEP and nighttime terrestrial ecosystem respiration, and temperature-fitted terrestrial ecosystem respiration during daytime. The response of NEP to flash droughts shows the compound effects of vegetation photosynthesis and ecosystem respiration. In terms of RT, the response of NEP is slower than GPP for SAV, but is quicker for DBF and ENF (Figure 5). The discrepancies between NEP and SM in terms of RTmin are more obvious than those between GPP and SM, and the RTmin of NEP is much shorter than the RTmin of soil moisture especially for DBF and ENF, which may be related to the increase of ecosystem respiration (Figures 5 i and j).

Figure 6 shows the temporal changes of SGPPA and soil moisture percentiles during 8 days before soil moisture flash droughts and during the first 24 days of the droughts. During 8 days before flash droughts, there is nearly no obvious decline for SGPPA, while SAV, DBF and ENF shows small increase in GPP. The decline in SGPPA is more significant during the first 9-24 days of soil moisture flash droughts for different vegetation types, and SGPPA for SAV and CROP show quicker decline even during the first 8 days of soil moisture flash droughts. The decline rates in soil moisture are mainly concentrated within the first 16 days of flash droughts. There are various lag times for the response of GPP to the decline in soil moisture among different vegetation." (L337-375)

[Figure]

**Figure 4.** Percentage of the response time (days) of the first occurrence of negative GPP anomaly (a-e), minimum GPP anomaly and minimum soil moisture percentile (f-j) during soil moisture flash drought for different vegetation types. SAV: savanna, CROP: rainfed cropland, MF: mixed forest, DBF: deciduous broadleaf forest and ENF: evergreen needleleaf forest.

[Figure]

**Figure 5.** The same as Figure 4, but for net ecosystem productivity (NEP).

[Figure]

**Figure 6.** The temporal change rates of standardized GPP anomalies (a-e) and soil moisture percentiles (f-j) for different vegetation types. SAV: savanna, CROP: rainfed cropland, MF: mixed forest, DBF: deciduous broadleaf forest and ENF: evergreen needleleaf forest.

*Specific Comments:*

*4) Line 37-Insert "future" before "land carbon uptake" in this sentence.*

*5) Line 58-Please add the Svoboda et al. (2002) reference for the U.S. Drought Monitor.*

**Response:** Revised as suggested. (L38; L60)

*6) Line 59-This drought also impacted parts of southern Canada.*

**Response:** We have revised it as:

"He et al. (2019) assessed the impacts of the 2017 northern USA flash drought (which also impacted parts of southern Canada) on vegetation productivity based on GOME-2 solar-induced fluorescence (SIF) and satellite-based evapotranspiration." (L61-64)

*7) Line 78-Few studies, or no studies have investigated this parameter? If there are previous studies, please cite them here.*

**Response:** We have revised as "…few studies have investigated WUE during flash droughts that usually occur at sub-seasonal time scale (Xie et al., 2016; Zhang et al., 2019)." (L83-85)

*8) Introduction-It would also be good to cite the Otkin et al. (2018; WCAS) paper because they examined the impact of a flash drought on vegetation health across the north-central U.S.*

**Response:** We have revised as:

"Besides, the 2016 flash drought over U.S. northern plains also decreased agricultural production (Otkin et al., 2018b)." (L67-68)

*9) Line 99-Please add some additional information about the soil moisture sensors, such as their type, their accuracy, and how they are sited. It would also be good to know what the soil type is for each of the stations.*

`Response:` We have added additional information as follows:

"Soil moisture observations are usually averaged over multiple sensors including time domain reflectometer (TDR), frequency domain reflectometer (FDR), and water content reflectometer etc. However, the older devices may be replaced with newer devices at certain sites, which may decrease the stability of long-term soil moisture observations and the average observation error of soil moisture is ±2%." (L106-111)

*10) Line 103-106 -How were these vegetation classifications determined? I think it would also be good to briefly discuss the phenological characteristics of these classifications.*

**Response:** We have clarified the classification as follows:

"The vegetation classification is according to International Geosphere-Biosphere Program (IGBP; Belward et al., 1999), where MF is dominated by neither deciduous nor evergreen tree types with tree cover larger than 60%, and the land tree cover is 10-30% for SAV." (L121-124)

*11) Line 106- Please make this sentence explicit rather than simply stating "etc". Also, this would be a good spot to point the readers to the top panel in Fig.2 to see the locations of these stations.*

**Response:** We have revised the manuscript as follows:

"Here we only select the FLUXNET observations including 12 evergreen needleleaf forest sites (ENF), 5 deciduous broadleaf forests (DBF), 6 crop sites (CROP; 5 rain-fed sites and 1 irrigated site), 3 mixed forests (MF), and 3 savannas (SAV). The sites for grasslands, evergreen broadleaf forests, and shrublands are excluded because there are less than 10 soil moisture flash drought events." (L116-121)
We have also revised Figure 2. Please see our response above.

*12) Lines 106-108-Please provide some justification for why these three particular sites were chosen for the case study analyses. It would also be helpful to mention here where these three stations are located, and a brief overview of their climate*

*characteristics. For example, are there stations located in regions that are known to frequently experience flash droughts?*

**Response:** We have removed the case analyses in the revised manuscript because they cannot represent the situations for different vegetation types. Instead, we have now focused on the composite analysis of soil moisture flash droughts for each vegetation type.

*13) Line 116-Does the first day of the flash drought occur at the beginning, middle, or end of the 8-day period used to compute the mean conditions? Please clarify.*

**Response:** We have clarified it as follows:

"1) Soil moisture flash drought starts at the middle day of the 8-day period when the 8-day mean soil moisture is less than the 40$^{th}$ percentile, and the 8-day mean soil moisture prior to the starting time should be higher than 40$^{th}$ percentile to ensure the transition from a non-drought condition." (L136-140)

*14) Figure 1-The label between steps 2 and 3 should be "true". The box for step five should also be expanded to include "and <2 months". Please correct these errors.*

**Response:** We have now removed the maximum duration threshold and updated the figure. Please see our response above.

*15) Line 119-It would be good to note here that these differences are also being computed at 8-day increments to match the cadence of the 8-day mean periods.*

**Response:** Thanks for your comments. We have revised the manuscript as follows:

"2) The mean decreasing rate of 8-day mean soil moisture percentile should be no less than 5% per 8 days to address the rapid drought intensification." (L140-141)

*16) Lines 123-125-"Recovery" is imprecise here because a decrease of 4% from one period to the next does not represent recovery; instead, it simply means that the deterioration is not fast enough to meet the threshold for a flash drought used in this study. Please change this term to "stabilization", or something similar, because that*

*will permit some degradation to still occur. Note that this only refers to the soil moisture status "stabilizing", thus, the inconsistency with respect to the vegetation parameters (see Major Comment#1) still remains and will also need to be properly addressed.*

**Response:** Thanks for your comments. The end of the onset stage of flash drought occurs when the **mean decline rate (from the beginning of flash drought)** is smaller than 5% in percentiles per 8 days, which would avoid such phenomenon that the soil moisture percentiles are still declining after the onset stage as much as possible. We compared the soil moisture percentiles during recovery stages and at the ending point of onset stages, and found that the soil moisture still declines at the rate of 2~3% in percentiles per 8 days only for 3% of flash drought events (Figure R1). Therefore, the soil moisture percentiles during the identified recovery stages increase as compared with the ending point of onset stages for most cases.

[Figure]

**Figure R1.** The frequency of soil moisture percentile changes between recovery stages and the ending point of onset stages.

*17) Line 132-Please change the start of this sentence to "At least decade long"*

**Response:** Revised as suggested. (L154)

*18) Line 132-140-It would be good to reiterate here that the percentiles themselves are still only computed over an 8-day period, but that the use of the surrounding 8-day periods are used to increase the sample size. These surrounding time periods though are certainly not completely independent, so please also comment on how much this approach does or does not increase the effective sample size when computing the percentiles*

**Response:** Thanks for your comments. Figure R2 shows the probability density function of soil moisture at different time based on the climatology solely from the target time of all observation years (a_clim) and the climatology consisting of the target time and 8 days before and after the target time of all observation years (b_clim). The b_clim is smoother than a_clim, indicating that the extended samples would decrease the uncertainty caused by certain extreme values. We have revised the manuscript as follows:

"Besides, the target 8-day soil moisture percentiles are only based on the target 8-day soil moisture in the context of the expanded samples." (L157-159)

[Figure]

**Figure R2.** The probability density function of soil moisture at Jun 2-9, Jun 10-Jul 17, Jun 18-Jun 25, Jun 26-Jul 2 based on the climatology solely from the target time during all observation years (black lines; a_clim) and the climatology from not only the target time but also 8 days before and after the target time from all observation years (red lines; b_clim).

*19) Lines 150-Please add the Crausbay et al. (2017) paper in BAMS that discusses ecological drought.*

**Response:** Revised as suggested. (L174)

*20) Line 154-You highlight an example with 19 years of data: however, most of the stations only have around 10 years of data. This is a short period for computing standard deviations. Please comment on how the short period of record will impact the anomalies and their subsequent use in this study.*

**Response:** Thanks for your comments. Here the standardized deviation of GPP are also based on at least 30-sample climatology, which is same as that of soil moisture percentiles as we mentioned above. We have revised the manuscript as follows:

"For instance, all Apr 1-8 during 1996-2014 would have a $\mu_{GPP}$ and a $\sigma_{GPP}$ based on a climatology same as soil moisture percentile calculation, which consists of March 24-31, Apr 1-8, and Apr 9-16 in all years, and Apr 9-16 would have another $\mu_{GPP}$ and another $\sigma_{GPP}$, and so on" (L179-182)

*21) Lines 154-157-The example provided in this sentence implies that ecological drought always happens one period after the flash drought first develops. Is that the true intention here? If not, please clarify this sentence. I would expect there to be more than a one period lag because in many situations, the vegetation roots will extend much deeper than the 10-cm topsoil layer used in this study to identify flash droughts, thereby allowing them to remain healthy despite a rapidly drying topsoil layer. This needs to be highlighted in this section – a flash drought in the top soil layer may not correspond to an ecological drought because of the depth of the roots.*

**Response:** Thanks for your comments. We have revised the manuscript as follows:

"Considering flash drought is identified through surface soil moisture due to the availability of FLUXNET data, vegetation with deeper roots may obtain water in deep soil and remain healthy during flash drought. The roots vary among different vegetation types and forests are assumed to have deeper roots than grasslands, which may influence the response to soil moisture flash droughts." (L189-193)

*22) Lines 150-162-It would be helpful if each of these indices were assigned separate names to be used in the results section.*

*23) Line 187-Please add "or equal to" before 24 days*

**Response:** Revised as suggested.

*24) Line 190-The station level average lengths are not helpful because many of the stations only have one or two events. It would be better to show the average length over all of the stations, or for all of the stations within a particular ecosystem type. Please do this in the revised text.*

**Response:** We have now shown all results based on different vegetation types instead of at station level. The manuscript has been revised as follows:

"Figure 2a shows the distribution of the 29 sites with different vegetation types, which are mainly distributed over North America and Europe. The number of soil moisture flash drought ranges from 13 to 70 events among different vegetation types. There are 12 ENF sites in this study, and the number of soil moisture flash droughts for ENF (70) is the most among all the vegetation types. The duration for flash drought events ranges from 24 days to several months. In some extreme cases, the flash droughts would develop into long-term droughts without enough rainfall to alleviate drought conditions. Mean durations of soil moisture flash droughts for different vegetation types range from around 30 days to 50 days (Figure 2c)." (L231-241)

*25) Lines 192-193-Is this sentence meant to imply that some stations may have multiple flash droughts because a single event is broken into two because of a rainfall event that temporally improves things? If so, please describe it as such, otherwise it is not clear what this sentence adds to the paper.*

**Response:** This sentence has been deleted as it is not relevant to the results.

*26) Line 192-What is meant by "variability of soil moisture"? Please describe this more clearly. Also, this really means variability of precipitation since it is the ultimate cause of the variability in soil moisture.*

**Response:** The relationship between frequency of flash droughts and variability of soil moisture is not significant, so we have now deleted this sentence.

*27) Figure 2-The panels on this figure are difficult to read. For example, the spatial heterogeneity briefly mentioned in the text is impossible to see in the top panel because most of the stations are crammed into central Europe or North America, and it is impossible to relate the results shown in the bottom panels to the map shown in the top panel. I suggest breaking this panel into separate panels for North America, Europe, and the other four stations individually, while still taking the same amount of space as the current panel. This will allow you to zoom into all of these regions and therefore more clearly show the spatial heterogeneity.*

**Response:** We have revised Figure 2 as suggested. Please see our response above.

*28) Lines 204-206-This sentence is imprecise. A decrease in ET will indeed limit the loss of soil moisture; however, it does not represent an alleviation of drought conditions. For one thing, soil moisture will still be decreasing in the absence of rainfall, albeit at a slower rate. Secondly, decreasing ET actually means that agricultural or ecological drought conditions are worsening. Please clarify this statement to account for these considerations.*

**Response:** We have revised the sentence as follows:

"ET starts to decrease during the recovery stage due to the limitation of water availability, and the decreasing ET also reflects the enhanced water stress for vegetation during the recovery stage." (L413-415)

*29) Lines 210-211-Please add some information describing where these stations are*

*located, and why these events were chosen for closer analysis.*

*30) Figure 4-Please change the top and bottom rows so that precipitation and temperature anomalies can be both positive and negative, otherwise, the analysis is incomplete since only one part of the anomaly time series can be shown.*

*31) Line 220-This statement is too strong because it is based on a single case study.*

*32) Line 228-Is there a reference that supports this statement? The variability in the time series for this station is very similar to the other two time series shown on Fig.4.*

*33) Lines 230-231-This statement is not supported by the bottom row of Fig. 4 where the ET anomalies for this savanna station are actually less severe than those for the forested site. Please fix this in the revised text.*

*34) Lines 212, 224, and 236-It would help if you pointed the reader toward the appropriate panels on Fig. 4 in the introductions to each of these paragraphs.*

**Response:** We have removed the case analyses in the revised manuscript and focused on the composite analysis of flash droughts for each vegetation type. Please see our response above.

*35) Figure 5-Please move the legend on panel a to panel b since that is where both these lines are shown.*

**Response:** We have reorganized the manuscript according to your comment 3.

*36) Line 252-It would be good to clarify that is "flash drought as determined by soil moisture reductions"*

**Response:** Revised as suggested.

*37) Line 279-Why "down to its normal conditions"? I assume this is a mistake since you've already shown in previous section that GPP anomalies become negative during a flash drought.*

**Response:** In this study, negative GPP anomalies did not occur during all flash drought events and GPP responded to 81% of flash droughts. We have clarified as follows:

"Here, we select 81% of soil moisture flash drought events with GPP declining down to its normal conditions to analyze the interactions between carbon and water fluxes, while GPP during the remaining 19% of soil moisture flash drought events may stay stable and is less influenced by drought conditions." (L382-385)

*38) Line 284-The ratio is reversed compared to that shown at line 172.*

**Response:** Here uWUE (GPP $\times \sqrt{VPD}/ET$) is partitioned into GPP and $ET/\sqrt{VPD}$, which is more direct when compared the response of vegetation photosynthesis and stomatal conductance to soil moisture flash droughts, respectively.

*39) Line 288-Again, this terminology is confusing-how can "recovery" be accompanied by "significant reductions" in GPP and ET. Those reductions show that vegetation conditions have deteriorated, not improved. This is also repeated at lines 319-320. This terminology needs to be changed to reflect that the "recovery" is only respect to soil moisture.*

**Response:** We highlight the recovery is referred to soil moisture flash droughts (L401).

*40) Line 315-Please change "intensify" to "reduction".*

**Response:** Revised as suggested.

**Response to the comments from Reviewer #2**

We are grateful to the reviewer for the constructive and careful review. The constructive suggestions have helped improved our manuscript. The reviewer's comments are italicized and our responses immediately follow.

*The authors present first evaluation of GPP from FLUXNET in response to flash drought. This is an important topic and this submission is timely as well as novel. At the same time, I feel that a more detailed analysis is warranted before publication. General comments:*

*1) I generally think that analyzing the relationships between flash drought and GPP is very important. I am wondering though, whether this paper leaves out a large part of the story by focusing narrowly on the 30-60 days of flash drought. Similarly, there is very little analysis that looks into the underlying mechanisms of GPP besides the WUE analysis. I am wondering how temperature, global radiation, SM, and VPD, which all affect GPP behave. For example one would expect drought to be associated with elevated temperatures. In this context, the authors stress the GPP reduction associated with drought, but several other papers have shown that GPP reduction during drought can be associated with compensation effects before and after the drought. By only focusing strictly on the drought these are being missed.*

**Response:** Thanks for your comments. In the revised manuscript, we have now dropped the maximum length threshold of 60 days for the definition of flash drought, although the main conclusions remain unchanged. To explore the role of climate factors on GPP, we have now used partial correlation to investigate the relationship between the standardized anomalies of GPP and temperature, radiation, VPD and soil moisture. Besides, we have extended the study period from 8 days before flash drought to 8 days after flash drought. There is little change of GPP during 8 days before flash droughts, and the decreasing in GPP is more obvious during the recovery stage of flash droughts and 8 days after. The deficits in soil moisture play an important role in decreasing GPP during onset stages of flash droughts, whereas VPD

is more significant to GPP during recovery stages. We have revised the manuscript as follows:

**"2.2.4 The role of meteorological conditions on GPP**

Considering the compound impacts of temperature, radiation, VPD and soil moisture on vegetation photosynthesis, the partial correlation is used to investigate the relationship between GPP and each climate factor, with the other 3 climate factors as control variables as follows:

$$r_{ij(m_1,m_2…m_n)} = \frac{r_{ij(m_1,…m_{n-1})} - r_{im_n(m_1,…m_{n-1})}r_{jm_n(m_1,…m_{n-1})}}{\sqrt{(1-r^2_{in(m_1,…m_{n-1})})(1-r^2_{jn(m_1,…m_{n-1})})}} \tag{1}$$

where $i$ represents GPP, $j$ represents the target meteorological variables and $m_1, m_{2…}$ and $m_n$ represent the control meteorological variables. $r_{ij(m_1,m_2…m_n)}$ is the partial correlation coefficient between $i$ and $j$, and $r_{ij(m_1,…m_{n-1})}$, $r_{im_n(m_1,…m_{n-1})}$ and $r_{jm_n(m_1,…m_{n-1})}$ are partial correlation coefficients between $i$ and $j$, $i$ and $m_n$, $j$ and $m_n$ respectively under control of $m_1, m_{2…}$ and $m_{n-1}$." (L215-226)

"**3.4 The role of climate factors on GPP during soil moisture flash drought**

Figure 8 shows the partial correlation coefficients between standardized anomalies of GPP and meteorological variables and soil moisture percentiles during different stages of soil moisture flash droughts. The correlation between climate factors and GPP is not statistically significant during 8 days before soil moisture flash droughts. During onset stages of soil moisture flash droughts, the partial correlation coefficients between SGPPA and soil moisture percentiles are 0.44, 0.49 and 0.29, respectively for SAV, CROP, and ENF (p<0.05). Besides, shortwave radiation is positively correlated with SGPPA for MF, DBF, and EBF (Figure 8b) during onset stages and the positive anomalies of shortwave radiation could partially offset the loss of vegetation photosynthesis due to the deficits in soil moisture. SGPP is also positively correlated with temperature during onset stages for SAV and DBF. The partial correlation coefficients between SGPPA and VPD are -0.53 and -0.22 respectively for DBF and ENF, and the higher VPD would further decrease GPP during onset stages. The influence of VPD on GPP is much more significant during recovery stages and 8 days after. SGPPA is positively correlated with soil moisture and negatively with VPD for SAV both during recovery stages and 8 days after." (L423-439)

"During 8 days before soil moisture flash drought, WUE and uWUE are generally close to the climatology (Figure 7a) and there are no significant changes in GPP, ET, and $ET/\sqrt{VPD}$ (Figures 7e and 7i). However, the median value of SGPPA for SAV is positive (Figure 7e)." (L385-389)

"During 8 days after flash drought, the standardized anomalies of uWUE are still positive for forests, whereas SGPPA and ET are both lower than the climatology for all ecosystems. The ecological negative effect would persist after the soil moisture flash drought." (L419-422)

[Figure]

**Figure 7.** Standardized anomalies of water use efficiency (WUE), underlying WUE (uWUE), GPP, ET and $ET/\sqrt{VPD}$ during 8 days before flash drought onset, onset and recovery stages of flash drought events, and 8 days after flash drought.

[Figure]

**Figure 8.** The partial correlation coefficients between GPP and soil moisture (SM), shortwave radiation (SW), temperature (Temp) and vapor pressure deficit (VPD) for different vegetation types including savannas (SAV), rain-fed croplands (CROP), mixed forests (MF), deciduous broadleaf forests (DBF), and evergreen needleleaf forests (ENF) during 8 days before soil moisture flash drought, onset and recovery stages and 8 days after soil moisture flash drought. * indicates the correlation is statistically significant at the 95% level. (L934-941)

*2) Similarly, the authors bin data based on onset (which should probably rather be called intensification) and recovery time as well as 8-day intervals. They present 3 examples of flash droughts in Figure 4, but it is unclear to me to what extent these are being representative and whether it makes sense to lump all drought events together like this. For example, the FI-Sod event shows fast recovery in SM, GPP, and ET (i.e. is terminated by a strong rain event), while US-SRM and IT-Col show basically no recovery of GPP and only ET recovery for IT-Col, which indicates that there is no real recovery taking place. Based on this, I would not expect to find generalizable*

*behavior during this period. I am not sure how to resolve this in detail, but I think that a deeper dive into data and individual events is merited.*

**Response:** Thanks for your comments. We have removed the case analysis in the revised manuscript because they cannot represent different vegetation types. Instead, we have now focused on the composite analysis for each vegetation type throughout the manuscript. This study focuses on the ecological response during the onset and recovery stages of flash droughts. However, it is still an important issue to assess the ecological impacts after flash droughts. Therefore, we use lagged autocorrelation models to investigate the relationship between GPP and soil moisture conditions during 8 days after flash droughts, and GPP at the end of flash droughts as follows:

$$GPP_{t+1} = b_0 + b_1 SM_{t+1} + b_2 GPP_t \tag{1}$$

where $GPP_{t+1}$ and $SM_{t+1}$ are the standardized anomalies of GPP and soil moisture percentiles during 8 days after flash droughts, and $GPP_t$ is the GPP at the end of flash droughts. $b_0$, $b_1$ and $b_2$ are empirically derived coefficients. Table R1 shows the regression coefficients of b1 and b2. The regression coefficients for soil moisture during 8 days after flash droughts is significantly positive for SAV, DBF, and ENF, and the regression coefficients for GPP at the end of flash droughts are also positive for SAV and CROP (Table R1). These indicate that the antecedent vegetation conditions and soil moisture after flash droughts would influence the GPP at different ecosystems.

**Table R1.** The regression coefficients of b1 and b2 for soil moisture during 8 days after flash droughts and the GPP at the end of flash droughts, respectively. * indicates statistically significant at the 95% level.

|     | SAV | CROP | MF | DBF | ENF |
| --- | --- | --- | --- | --- | --- |
| b1 | 0.009* | -0.006 | -0.006 | 0.007* | 0.001* |
| b2 | 0.82* | 0.52* | 0.11 | 0.61 | 0.56 |

Thus, we have added the discussion about the legacy effects of flash droughts connected with climate and vegetation conditions in the revised manuscript as follows:

"During 8 days after the soil moisture flash drought, the anomalies of GPP and ET are still negative, indicating that the vegetation does not recover immediately after the soil moisture flash drought. The legacy effects of flash droughts may be related to the vegetation and climate conditions (Barnes et al., 2016; Kannenberg et al., 2020)." (L479-483 in the revised manuscript)

*3) The discussion is falling a bit short with respect to differences between plant functional type classes. Some discussion around differences between grasslands and forests as outlined in specific comments may help here.*

**Response:** Thanks for your comments. We have compared the response of NEP and GPP and discussed the correlation between soil moisture and GPP for different vegetation types in the revised manuscript as follows:

"Due to the influence of ecosystem respiration, the responses of NEP for DBF and ENF to flash droughts are much quicker than GPP, implying that the sensitivity of ecosystem respiration is less than that of vegetation photosynthesis (Granier et al., 2007)." (L457-460)

 "Due to the limitation of FLUXNET soil moisture measurements, here we used soil moisture observations mainly at the depths of 5 to 10 cm. We also analyzed the response of GPP to flash drought identified by 0.25-degree ERA5 soil moisture reanalysis data at the depths of 7cm and 1m. The response of GPP to flash droughts identified by FLUXNET surface soil moisture are quite similar to those identified by ERA5 soil moisture at the depth of 1m (not shown). There are less GPP responses to flash droughts identified by ERA5 surface soil moisture. Although we select the ERA5 grid cell that is closest to the FLUXNET site and use the ERA5 soil moisture data over the same period as the FLUXNET data, we should acknowledge that the gridded ERA5 data might not be able to represent the soil moisture conditions as well as flash droughts at in-situ scale due to strong heterogeneity of land surface. Therefore, the in-situ surface soil moisture from FLUXNET is useful to identify flash droughts compared with reanalysis soil moisture, although the in-situ root-zone soil moisture would be better." (L490-504)

"The correlation between soil moisture and GPP is more significant for SAV, CROP, and ENF during onset stages of flash droughts, which is consistent with the strong response to water availability of SAV and CROP (Gerken et al., 2019). SAV is more isohydric than forests and would reduce stomatal conductance immediately to prohibit water loss that further exacerbates drought (Novick et al., 2016; Roman et al., 2015). However, almost all vegetation types show high sensitivity to VPD during the recovery stage of flash droughts." (L519-525)

*4)Given that FLUXNET measures NEE rather than GPP and GPP is partitioned, some discussion on this partitioning may be warranted and NEE should probably also be shown.*

**Response:** Thanks for your positive comments. We have clarified the measurement of NEP and revised our manuscript as follows:

"GPP is derived from direct eddy covariance observations of NEP and nighttime terrestrial ecosystem respiration, and temperature-fitted terrestrial ecosystem respiration during daytime. The response of NEP to flash droughts shows the compound effects of vegetation photosynthesis and ecosystem respiration. In terms of RT, the response of NEP is slower than GPP for SAV, but is quicker for DBF and ENF (Figure 5). The discrepancies between NEP and SM in terms of RTmin are more obvious than those between GPP and SM, and the RTmin of NEP is much shorter than the RTmin of soil moisture especially for DBF and ENF, which may be related to the increase of ecosystem respiration (Figures 5 i and j)." (L355-364)

"Due to the influence of ecosystem respiration, the responses of NEP for DBF and ENF to flash droughts are much quicker than GPP, implying that the sensitivity of ecosystem respiration is less than that of vegetation photosynthesis (Granier et al., 2007)." (L457-460)

[Figure]

**Figure 5.** The same as Figure 4, but for net ecosystem productivity (NEP).

*Specific Comments:*

*L99: It might be a good idea to also look into other sources of soil moisture here, as there is little standardization across FLUXNET with respect to sensor depth etc.*

**Response:** Here we used 0.25-degree ERA5 soil moisture reanalysis data at the depths of 7cm and 1m to analyze the response of GPP to soil moisture flash droughts and added this into discussion in the response to your comment 3.

*L101: We select 34 sites from FLUXNET where,…>are these all sites that fit the definition from this sentence or was there further subsetting done?*

**Response:** We have clarified as follows:

"Here we only select the FLUXNET observations including 12 evergreen needleleaf forest sites (ENF), 5 deciduous broadleaf forests (DBF), 6 crop sites (CROP; 5 rain-fed sites and 1 irrigated site), 3 mixed forests (MF), and 3 savannas (SAV). The sites for grasslands, evergreen broadleaf forests, and shrublands are excluded because there are less than 10 soil moisture flash drought events." (L116-121)

*L147: "The negative anomalies of GPP during flash drought are considered as the signal of ecological deterioration."> This sounds not correct to me. Water stress will reduce GPP, which is a given, but I don't think it necessarily follows that this has a lasting consequence as implies here. It would be interesting to see to what extent do these ecosystems compensate. I.e. is there a lasting effect from a flash drought even in the annual carbon balance.*

**Response:** We agree with the reviewer that a GPP decline below its normal condition (long-term mean) does not necessarily indicate an ecological deterioration, where we actually regard it as the onset of ecological response. We have examined the GPP response during 8 days after flash droughts (please see our response to your first comment) and we have revised this sentence as follows:

"The negative anomalies of GPP during soil moisture flash drought are considered as the onset of ecological response." (L171-173)

*L165: "influence of water and energy conditions">" water and energy availability?"*

**Response:** Revised as suggested.

*L189-190: "and the mean durations were from around 30 days to 60 days among FLUXNET sites"> I am a bit confused by that given that I was under the impression that droughts longer than 2 months days were excluded from the analysis. How can then mean drought length be 60 days, if that is also about the maximum possible length?*

**Response:** In the revised manuscript, we have now removed the threshold of maximum duration of flash droughts and the average duration is calculated for each vegetation type not for each site.

*Figure 2 is problematic: I would zoom into Europe. It is also not possible to link the sites from a) to b) and c) without consulting Table 1. As a side note: the 4 Canadian ENF sites are more or less directly adjacent to each other, with 3 of them showing almost the same behavior. It may be better to only keep two of them (CA-TP4 is different (Why?))*

**Response:** Thanks for your comments. There are 4 Canadian ENF sites including CA-Obs, CA-TP1, CA-TP3, and CA-TP4 in this study. Although the vegetation type and climate conditions are quite similar for CA-TP1, CA-TP3, and CA-TP4, the ages of trees are different, which may influence soil moisture conditions and the ecological response to soil moisture flash droughts. We have revised Figure 2 as follows:

[Figure]

**Figure 2.** (a) Global maps of 29 FLUXNET sites used in this study. (b) Total numbers (events) and (c) mean durations (days) of soil moisture flash drought events for each vegetation type during their corresponding periods (see Table 1 for details). Different colors represent different vegetation types.

*Figure 3 and associated text: I am a bit confused about onset and recovery. Are these singe 8 day periods or do they refer to several periods. I am not sure whether this is necessarily a good way of showing this data and what is really learned here, since everything is lumped together and there is an implied time-axis, which is not*

*consistent in itself. The temporal evolution of these events is also already well established in the literature.*

**Response:** To clarify different stages in the Figure 3, we have revised manuscript as follows:

"Here the onset and recovery stages of flash droughts refer to certain periods characterized by the soil moisture decline rates. The standardized anomalies of temperature, precipitation, VPD, and shortwave and soil moisture percentiles are composited to show the meteorological conditions during different stages of flash droughts." (L250-254)

*Figure 4: It looks as if these sites were chosen as representative for each class, but this should be made explicit in the text. I don't particularly like the fact that anomalies are being plotted at the site level. We need to calculate ET, GPP, and SM anomalies to compare sites and establish drought, but here there is no need and it makes it harder to understand the underlying dynamics. I also think that if these sites are chosen, one should plot all drought events (all six or so per site) and not only specifically chosen year. Also, based on this figure, I feel that onset should be renamed as intensification.*

**Response:** Thanks for your comment. We have removed the case analysis in the revised manuscript because they cannot represent different vegetation types. Instead, we have now focused on the composite analysis for each vegetation type throughout the manuscript. "Intensification" and "onset" are quite similar to describe the development of flash droughts. Corresponding with "recovery", "onset" would be a better name than "intensification" (Yuan et al., 2019).

*Figure 5: a) It appears if there is a quick response of GPP at the beginning of the flash drought, which one would expect simply by having high VPD, which will lead to stomata closure, but SM seems to be much less affected. It would be nice to learn whether this is really unusual or whether this GPP responses related to soil moisture reduction (drought) or VPD forcing. For example Gerken et al. 2018*

*(https://www.hydrolearth-syst-sci-discuss.net/hess-2018-211/) showed that potential evapotranspiration (~VPD) happened before the onset 2017 Norther Great Plains flash drought. It would be interesting to see whether GPP reduction also occurs before drought onset. To what extent are panels c and d necessary.*

**Response:** Thanks for your positive comments. We analyzed the standardized GPP anomalies during 8 days before flash drought and there is no obvious decline in GPP except for MF (Figure 6e). Besides, the decline in soil moisture plays a dominant role in affecting GPP during onset stages of flash droughts and the influence of higher VPD is more significant during recovery stages. Please see our response to your first comment.

*L251: "that negative GPP anomalies occur during 81%"-> if this refers to the rad line in Figure 5a/b, then this number seems inconsistent with the figure, where it is more like 78%.*

**Response:** In the last version of manuscript, Figures 5a and 5b only showed the cumulative response frequency within 1-40 days of flash droughts, whereas the total response frequency is 81% during the whole flash droughts. In the revised manuscript, we have deleted Figure 5 and focused on ecological responses to flash droughts for different ecosystems.

*L270: "The result is consistent with the high vulnerability of vegetation in semiarid regions" > I would caution against this interpretation. Semi-arid ecosystems are highly adapted to changes in water availability and show fast response to changes in water availability (e.g. Gerken et al. 2019, 10.1038/s41612-019-0094-4). Without additional analysis, this should not be taken as a sign of degradation or vulnerability; especially since the final cumulative values are practically the same as for forests (MF, BF, ENF). Some discussion about isohydricity, VPD may also be helpful in this context (e.g. Novick et al, 2016, 10.1038/nclimate3114, Roman et al, 2015; 10.1007/s00442015-3380-9)*

**Response:** Although the final cumulative values are similar to those for forests, GPP for Savanna does show faster response to flash drought as illustrated in Figure 4 in the revised manuscript. However, we agree with the reviewer that the statement of "high vulnerability of vegetation in semiarid regions" is not relevant. We have revised the manuscript as follows:

"The result is consistent previous studies regarding the strong response of semi-arid ecosystems to water availability (Gerken et al., 2019; Vicente-Serrano et al., 2013; Zeng et al., 2018) and the decline in GPP for SAV is more related to isohydric behaviors during soil moisture drought and higher VPD, through closing stomata to decrease water loss as transpiration and carbon assimilation (Novick et al., 2016; Roman et al., 2015)." (L342-348)

*L285: "Increasing VPD and deficits in soil moisture would decrease canopy conductance" -> The fact that uWUE stays invariant shows that GPP reductions are due to canopy conductance. During recovery SAV and CROP, which are both dominated by grasses are likely brown, while forests are still green and quickly respond. This again likes directly to different biophysical responses of forests and grasslands and isohydricity effects. These should be discussed.*

**Response:** Thanks for your constructive comments. We have incorporated them into the revised manuscript as follows:

"The decrease in uWUE for SAV and CROP during recovery stages indicates that SAV and CROP are likely brown due to carbon starvation caused by the significant decrease in stomatal conductance (McDowell et al., 2008)." (L405-408)

"However, the positive anomalies of uWUE for DBF and ENF during the recovery stage imply that the decline in GPP mainly results from the stomata closure." (L411-413)

*L315: "Eventually, 81% of flash drought events cause negative ecological impacts on GPP." > I am not sure that a reduction in GPP is necessarily an negative impact. This depends greatly on the annual carbon balance. For example Wolf et al, 2016*

*(PNAS) showed that there is GPP compensation (i.e. warmer temperatures before drought causes higher initial GPP). Without looking into potential compensation effects, I feel that this statement is too harsh.*

**Response:** Thanks for your comments. We explored the response of GPP during 8 days before and after flash droughts and their relationship with soil moisture conditions and antecedent vegetation conditions, and found that there is no obvious anomaly in GPP during 8 days before flash droughts but GPP does not recover immediately as the end of flash droughts, and the legacy effects of soil moisture flash droughts on vegetation may be related to soil moisture conditions after flash droughts and the intensity of GPP response (please see our responses to your first two comments). Besides, we have revised the statement as follows:

"Eventually, 81% of soil moisture flash drought events cause declines in GPP." (L460-461)

*L346: "The positive anomalies of WUE and uWUE for forests show the adaptation of vegetation to flash drought from physiological perspective." > Not sure that this is true. Forests have also access to more water in the soil due to deeper roots and have invested much more in biomass. Grasslands just become dry and then recover. I think that these are different strategies rather than one being more prepared than the other.*

**Response:** Thanks for your comments. We have revised the manuscript as follows:

"The positive anomalies of WUE and uWUE for forests suggest that their deeper roots can obtain more water than grasslands during flash drought." (L512-515)

*Technical (not complete): L36: (e.g. droughtS, heat waveS)*

*L40: in some -> during (some is also not needed because of can)*

*L269: impaired -> reduced*

**Response:** Revised as suggested. (L37; L41; L343)

[revised manuscript text omitted]

---

## Author Response (AR2)

Xing Yuan
Professor
School of Hydrology and Water Resources
Nanjing University of Information Science and Technology
 No.219, Ningliu Road, Nanjing 210044, Jiangsu, China
Email: xyuan@nuist.edu.cn
Tel: +86-025-58699958
 https://orcid.org/0000-0001-6983-7368
September 25, 2020

Re: hess-2020-185

Dear Dr. Hildebrandt,

Regarding your decision letter on our manuscript entitled "Rapid reduction in ecosystem productivity caused by flash drought based on decade-long FLUXNET observations" (hess-2020-185), we have now carefully considered Reviewer #2's comments and incorporated them into the manuscript to the extent possible. We hope that you find the revised manuscript and the response acceptable to *Hydrology and Earth System Sciences*. The detailed responses to the comments are attached.

We appreciate the effort you spent to process the manuscript and look forward to hearing from you soon.

Sincerely yours,

Xing Yuan

**Response to the comments from Reviewer #2**

We are grateful to the reviewer for the careful review. We have incorporated the comments to the extent possible. The reviewer's comments are italicized and our responses immediately follow.

*General Comments:*

*The authors should revise the introduction to make clear the differences between flash drought, drought, and ecological drought. For the less informed reader, it may still be a bit confusing:*

*- Flash drought is the rapid intensification portion*

*- (meteorological or SM) drought is the prolonged reduction of SM*

*- ecological drought is directly associated with ecological effects such as GPP.*

*In the same light, I am still a bit confused to what extent this is specific to flash drought. Drought can develop into ecological drought, but it would be nice to have a discussion on what is special about flash droughts in the context of GPP as opposed to 'normal' drought.*

**Response:** Thanks for the reviewer's comments. For flash drought, we both consider the rapid onset phase and the recovery stage. We have clarified the difference between flash droughts and traditional droughts, and the ecological impacts of flash droughts in the revised manuscript as follows:

"Flash drought may only need several weeks to develop into its maximum intensity, and the rapid onset distinguishes it from traditional drought which is assumed to be a slowly evolving climate phenomenon taking several months or even years to develop (Otkin et al., 2018). Several extreme flash droughts would ultimately propagate into long-term droughts due to persistent precipitation deficits, e.g., 2012 flash drought over the USA Midwest Plain (Basara et al., 2019)." (L50-55)

"Here we consider not only the rapid onset stage of soil moisture flash droughts but also the recovery stage to assess the ecological impacts. The ecological responses to water stress vary under different ecosystems and drought characteristics, and the focus on the soil moisture flash droughts would detect the breakdown of ecosystem functioning of photosynthesis." (L98-102)

"Previous studies mainly focused on the response of vegetation to long-term droughts, and found that the response time ranged from several months to years through correlation analysis (Vicente-Serrano et al., 2013; Xu et al., 2018). The response time of vegetation to flash droughts might be different, which requires further investigation for quantification." (L62-66)

*Specific Comments:*

*1) L190-194: this sounds a bit informal and may need a citation*

**Response:** Thanks for the reviewer's comment. We have revised the manuscript as follows:

"Carbon assimilation and transpiration are coupled by stomates, and plants face a tradeoff between carbon uptake through photosynthesis and water loss through transpiration under the influence of water and energy availability (Boese et al., 2019; Gentine et al., 2019; Huang et al., 2016; Nelson et al., 2018). WUE can be used to quantify the trade-off between carbon and water cycles, which is defined as the assimilated amount of carbon per unit of water loss (Peters et al., 2018)." (L204-211)

*2) L239: the authors should clarify that flash drought refers to the initial intensification and that flash droughts can lead to prolonged meteorological drought with ecological consequences*

**Response:** The unique nature of flash drought is rapid intensification, and it may or may not develop into prolonged drought. We have clarified as follows:

"The onset stage of soil moisture flash droughts mainly refers to the rapid intensification, and the flash droughts may or may not develop into long-term droughts depending on the deficits in precipitation." (L256-259)

*3) Section 3.4: The partial correlation analysis. I am a bit confused by the*

*anti-correlation with VPD, given that VPD (or basically evaporative stress) has been found to be a major driver for flash drought.*

*This section should also be discussion in the discussion section, because I am not sure what this means.*

**Response:** The partial correlation is used to analyze the climate controls on GPP during different phases of soil moisture flash droughts. The negative correlation between VPD and GPP is found during onset and recovery stages of soil moisture flash droughts, and higher VPD would further decrease stomatal conductance thus decreasing carbon uptake. We have revised the manuscript as follows:

[revised manuscript text omitted]